# SplatFont3D: Structure-Aware Text-to-3D Artistic Font Generation with Part-Level Style Control

## Abstract

Artistic font generation (AFG) can assist human designers in creating innovative artistic fonts. However, most previous studies primarily focus on 2D artistic fonts in flat design, leaving personalized 3D-AFG largely underexplored. 3D-AFG not only enables applications in immersive 3D environments such as video games and animations, but also may enhance 2D-AFG by rendering 2D fonts of novel views. Moreover, unlike general 3D objects, 3D fonts exhibit precise semantics with strong structural constraints and also demand fine-grained part-level style control. To address these challenges, we propose SplatFont3D, a novel structure-aware text-to-3D AFG framework with 3D Gaussian splatting, which enables the creation of 3D artistic fonts from diverse style text prompts with precise part-level style control. Specifically, we first introduce a Glyph2Cloud module, which progressively enhances both the shapes and styles of 2D glyphs (or components) and produces their corresponding 3D point clouds for Gaussian initialization. The initialized 3D Gaussians are further optimized through interaction with a pretrained 2D diffusion model using score distillation sampling. To enable part-level control, we present a dynamic component assignment strategy that exploits the geometric priors of 3D Gaussians to partition components, while alleviating drift-induced entanglement during 3D Gaussian optimization. Our SplatFont3D provides more explicit and effective part-level style control than NeRF, attaining faster rendering efficiency. Experiments show that our SplatFont3D outperforms existing 3D models for 3D-AFG in style–text consistency, visual quality, and rendering efficiency.

## 1 Introduction

Artistic fonts are widely used in movie posters, brand icons, video games, and many other areas in our daily lives. Different from standard printed fonts in books and computers, artistic fonts attain significant diversity in glyph shapes and font effects. It generally requires expert human designers to create personalized artistic fonts depending on specific scenarios or contexts, which is highly demanding in both time and financial cost. Therefore, there is a pressing need for methods of Artistic Font Generation (AFG), which can teach machines to automatically generate artistic fonts. Such innovative techniques are supposed to assist human designers in creating customized 3D artistic fonts. With recent advances in GANs and diffusion models (Goodfellow et al., 2020; Ho et al., 2020), AFG has achieved remarkable success (Hayashi et al., 2019; Wang et al., 2023a; Li et al., 2023b; Miao et al., 2024; Mu et al., 2024; Ren et al., 2025).

Although previous studies are capable of generating novel 2D collections by combining various existing glyphs and textures, they are primarily limited to 2D artistic fonts in flat design, leaving 3D font synthesis largely underexplored. Compared with 2D-AFG, 3D-AFG offers broader application prospects and greater practical values. For example, most 2D-AFG methods are confined to creating 2D planar images from a pre-defined viewpoint but are not capable of generating novel views, limiting their flexibility and practicability. Instead, 3D artistic fonts can explicitly represent the spatial structures of fonts and can feasibly render 2D fonts of arbitrary views, thereby positioning 2D-AFG as a special case of its 3D counterpart. Moreover, 3D-AFG enables applications in immersive 3D environments such as 3D animations, video games, and virtual reality. Therefore, 3D-AFG exhibits significantly better application potential than 2D approaches, which is worth further investigation.

Nevertheless, 3D-AFG poses unique challenges beyond the general 3D object synthesis, essentially making existing text-to-3D models inapplicable. Specifically,

1. **Semantic & Style Constraints.** Different from general objects, character fonts encode rich semantic information, and their shapes are strictly constrained to preserve semantic correctness. However, existing pre-trained text-to-3D models (Lin et al., 2023; Wang et al., 2023b; Chen et al., 2024b; Huang et al., 2024; Liu et al., 2023b;a) or 2D diffusion models (Ho et al., 2020; Song et al., 2020; Rombach et al., 2022; Nichol et al., 2021) are primarily exposed to general objects, making them struggle with font recognition and understanding. This makes 3D-AFG particularly challenging, especially in cases that require both preserving correct semantics under shape constraints and incorporating accurate stylistic attributes at precise layout positions.

2. **Part-level Style Control.** A more practical 3D-AFG model should go beyond the global stylization and further achieve structure-aware synthesis with part-level control. However, part-level modification is considerably difficult for existing 3D models, highlighting their limitations for structure-aware 3D-AFG. For example, NeRF (Mildenhall et al., 2021) represents objects implicitly using a neural field that essentially lacks natural decomposition, thus making the part modifications of 3D objects difficult. Moreover, 3DGS (Kerbl et al., 2023) represents objects with 3D Gaussian points, but it carries no precise semantics for reliable component partitioning.

3. **Expensive Acquisition Cost.** Unlike 2D images, 3D fonts are considerably scarce and not feasibly obtainable from publicly available sources (e.g., the Internet). Furthermore, the creation and collection of 3D artistic fonts present significant challenges, as they demand that designers possess formal expertise in artistic font design and mastery of 3D modeling software (e.g., Maya and 3ds Max). This substantially increases the time and financial cost as well as the complexity of dataset creation. The scarcity of 3D font datasets eventually makes it impractical to obtain a generalized, large-scale 3D-AFG model through conventional supervised training.

So far, structure-aware 3D-AFG remains largely unexplored, and its challenges are still far from being addressed. Specifically, existing supervised text-to-3D approaches (Lin et al., 2023; Wang et al., 2023b; Chen et al., 2023a; 2024b; Metzer et al., 2023; Huang et al., 2024; Liu et al., 2023b;a) fail to address these challenges effectively due to the scarcity of 3D font data, which prevents building a general-purpose, highly generalizable 3D-AFG model. An alternative approach is to leverage large pre-trained 2D diffusion models for 3D generation (Poole et al., 2023; Lin et al., 2023; Wang et al., 2023b; Chen et al., 2023a; Metzer et al., 2023; Shi et al., 2023; Yi et al., 2024a; Chen et al., 2024b), thereby avoiding the need for extensive data collection. However, significant challenges remain for part-level 3D-AFG due to the implicit representation of NeRF (e.g., DreamFont3D (Li et al., 2024)) and the ambiguous component portioning for 3DGS (e.g., GaussianDreamer (Yi et al., 2024a)).

To address those challenges, this paper proposes SplatFont3D, a novel structure-aware text-to-3D artistic font generation model with precise part-level style control, which leverages the geometric advantages of 3DGS and the strong prior knowledge of pre-trained 2D diffusion models. Specifically, **(1)** We first introduce a Glyph2Cloud module to progressively refine the geometry shapes of 2D glyphs (or components) while maintaining their semantics and further construct well-initialized 3D point clouds for Gaussian initialization. **(2)** The initialized 3D Gaussians further leverage the priors of 2D diffusion models and are accumulatively optimized via Score Distillation Sampling (SDS), which projects the differentiable 3D representation from various viewpoints and makes the projected 2D images match the text conditions. Such a strategy eliminates the need for acquiring real 3D artistic font data, effectively addressing the challenges posed by data scarcity. **(3)** To achieve part-level style control, we exploit the geometric priors of 3D Gaussians to partition components for individual rendering. However, the dynamic optimization of 3DGS often causes Gaussian points to drift, and thus, points from different components may overlap and interfere with each other, ultimately degrading the generation quality. Hence, we further integrate a dynamic component assignment strategy to address this drift-induced component entanglement issue. This eventually enables more explicit and effective part-level style control than NeRF with faster rendering speeds. Experiments empirically demonstrate that our SplatFont3D can render 3D artistic fonts more effectively and efficiently than NeRF and existing text-to-3D models. Our contributions are summarized as follows:

- We propose SplatFont3D, a novel structure-aware text-to-3D artistic font generation model with precise part-level style control, a problem that has remained largely unexplored.

- We introduce Glyph2Cloud, a module that progressively refines the geometric shapes of 2D glyphs while maintaining original semantics, and consequently constructs well-initialized 3D point clouds for Gaussian initialization. This strategy enables the effective combination of 3DGS and 2D diffusion priors for 3D-AFG and further helps eliminate the need for acquiring real 3D font data, making the overall approach feasible.

- To enable precise part-level style control, we present dynamic component assignment that exploits the geometric priors of 3D Gaussians to partition components, while alleviating drift-induced entanglement during Gaussian optimization. Our explicit part-level control is more effective than the implicit one of NeRF, while attaining higher rendering efficiency.

- Extensive experiments demonstrate the superiority of our SplatFont3D over existing tex-to-3D models for 3D-AFG in style–text consistency, visual quality, and rendering efficiency.

## 2 RELATED WORK

### 2.1 ARTISTIC FONT GENERATION

Artistic font generation (Gao et al., 2019; Li et al., 2022; Wang et al., 2023a) has emerged as a vibrant research area. Early studies approached AFG via conditional GANs (Goodfellow et al., 2020), such as zi2zi (Tian, 2017) and GlyphGAN (Hayashi et al., 2019), which enabled style-consistent transfer across character sets. Subsequent research explored component-aware and few-shot paradigms for capturing better intra-character structures. Chen et al. (2024a) explicitly modeled ideographic composition for characters, and Li et al. (2023b); Park et al. (2021) employed the attention and global–local disentanglement to synthesize characters from only a few exemplars. With the advent of large-scale generative models, diffusion-based approaches have emerged. Wang et al. (2023a) demonstrated a pretrained text-to-image diffusion can be adapted to artistic typography, and Yang et al. (2023) further leveraged glyph shapes to balance creativity with legibility.

Nevertheless, most previous studies are confined to 2D rendering, leaving 3D artistic font generation largely underexplored. Prior attempt DreamFont3D (Li et al., 2024) tried to leverage pre-trained 2D diffusion models to refine 3D NeRF volumes. However, Such a NeRF model struggles to achieve precise part-level control, as the implicit representation of NeRF lacks structural decomposition. Moreover, the optimization and rendering of NeRF are highly time-consuming and computationally expensive. Our work differs by leveraging the strong geometry priors of 3D Gaussians and successfully achieves structure-aware personalized 3D-AFG with explicit part-level style control. Our SplatFont3D attains much higher rendering efficiency with better generation quality over Dream-Font3D, and our explicit part-level control is also more effective than the implicit one of NeRF.

### 2.2 TEXT-TO-3D GENERATION

Early text-to-3D methods (Chen et al., 2018; Seo et al.) mainly relied on large-scale 3D assets, which are limited by dataset coverage and struggle with novel shapes. Recent advances (Lin et al., 2023) leveraged Score Distillation Sampling (SDS) (Poole et al., 2023) to optimize NeRF (Mildenhall et al., 2021) with pretrained 2D diffusion models, eliminating the need for real 3D data. Moreover, point- and Gaussian-based representations (Kerbl et al., 2023; Yi et al., 2024a; Chen et al., 2024b) have been proposed to improve optimization speed, memory efficiency, and structure control. However, existing text-to-3D models are primarily designed for general 3D objects, essentially making those models inapplicable due to the unique challenges of fonts (such as the strong semantics and shape constraints of characters). Nevertheless, 3D-AFG via text-to-3D models remains largely underexplored, especially for structure-aware 3D generation with part-level style control.

### 2.3 3D EDITING

Besides text-to-3D generation, 3D editing has also witnessed significant progress in recent years, such as GaussianEditor (Wang et al., 2024), Control3D (Chen et al., 2023b), 3DStyleGLIP (Chung et al., 2024), SketchDream (Liu et al., 2024), and TIP-Editor (Zhuang et al., 2024). Different from 3D generation from scratch, 3D editing focuses on modifying an existing 3D model. Although 3D editing with local controls is feasible for 3D-AFG, direct 3D generation from scratch is more effective and efficient. The reasons include: (1) 3D generation with joint optimization integrates

both global and local objectives in a single pass, avoiding redundant two-pass 3D optimization of 3D editing. This results in faster rendering speed and lower I/O overhead than 3D editing. (2) 3D generation from scratch reduces geometric collapse and artifacts in edited regions and yields better global consistency than 3D editing, as local styles are regularized throughout the entire optimization process; (3) 3D editing requires precisely localizing part regions from 3D assets, however, which is difficult, especially for NeRF-based models and finer-grained editing on multiple components.

## 3 METHODOLOGY

### 3.1 OVERALL FRAMEWORK

As shown in Fig. 1, our SplatFont3D aims to generate 3D customized artistic fonts with part-level style control, which consists of three modules: (1) Glyph2Cloud (G2C) for 3D Gaussian initialization, (2) 3D Gaussians optimization via SDS, and (3) Dynamic Component Assignment (DCA) for part-level style control. It is worth noting that our SplatFont3D can also achieve "Global Style Generation", which is a simplified version of "Part-Level Control" that uses only a single component.

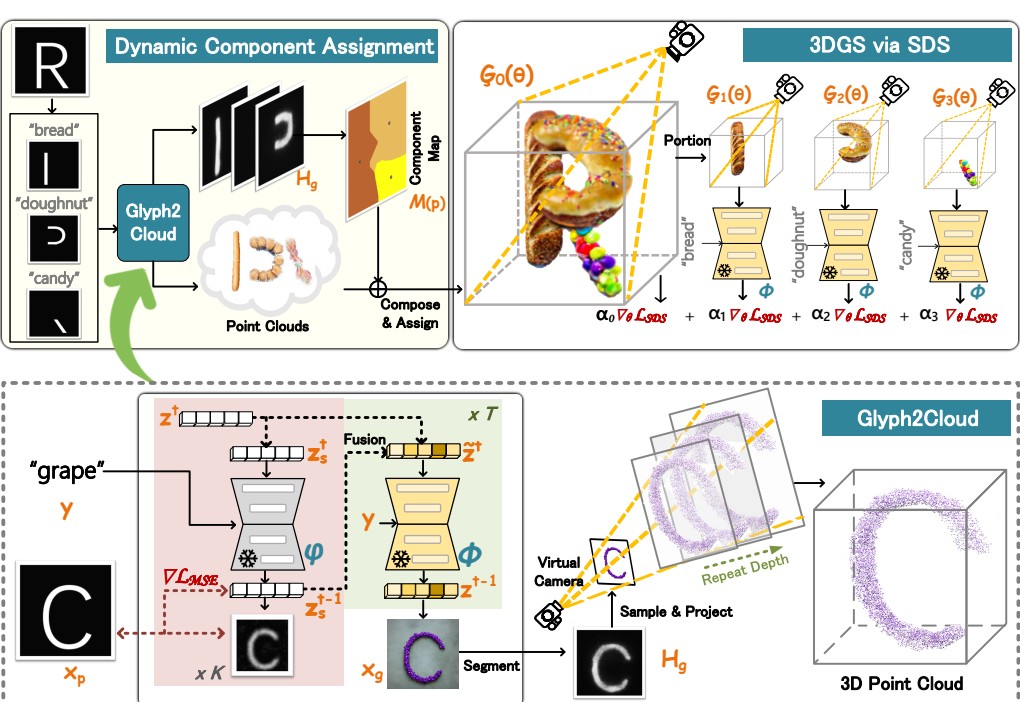

Figure 1: Overview of SplatFont3D for structure-aware 3D-AFG with part-level style control.

### 3.2 GLYPH2CLOUD FOR 3D GAUSSIAN INITIALIZATION

3DGS requires a well-initialized 3D point cloud for effective optimization, while it is challenging to directly generate accurate 3D font point clouds with pre-trained text-to-3D models. This is because these models are primarily trained on general objects and thus struggle with 3D font generation.

To address this issue, we propose **Glyph2Cloud (G2C)** to enable the creation of well-initialized 3D point cloud. The core idea is to leverage 2D printed glyphs as strong geometric priors and then utilize large pre-trained 2D diffusion models to generate 2D stylistic fonts that not only respect shape constraints but also preserve styles specified by textual prompts. After that, we can uniformly sample 2D pixels on the foreground region, and finally project these sampled 2D points into 3D space, thus obtaining the initial 3D point cloud for 3D Gaussian initialization.

**2D Generation with Shape-Style Tradeoffs.** We first adopt a pre-trained text-to-image diffusion model $\varphi$ to reconstruct the object shapes of input images in latent space. Let $z_t$ be the randomly

sampled latent noise, and $y$ be the text prompts that specify the artistic styles. Then the shape latent $z_s$ is dynamically calculated by reconstructing the input font mask $x_p$, i.e.,

$$z_s^t = z^t, \tag{1}$$
$$z_s^{t-1} = \varphi(z_s^t, y, t), \tag{2}$$

under which the shape latent $z_s^{t-1}$ is regularized through the shape constraints as

$$\mathcal{L}_{MSE} = ||\mathcal{D}(z_s^{t-1}) - x_p||_1, \tag{3}$$

where $\mathcal{D}$ is the pretrained image decoder. After that, we perform a denoising intervention by injecting the shape latent $z_s$ into the original latent $z_t$, which influences the generation of target 2D artistic fonts $x_g$ by enabling a trade-off between stylistic fidelity and shape preservation, i.e.,

$$\tilde{z}^t = \alpha \odot z_s^{t-1} + (1 - \alpha) \odot z^t \quad , \quad t = T, \ldots, T - K \tag{4}$$
$$z^{t-1} = \phi(\tilde{z}^t, y, t) \quad , \quad t = T, \ldots, 0 \tag{5}$$
$$x_g = \mathcal{D}(z^0), \tag{6}$$

where $\phi$ is a pre-trained text-to-image diffusion model.

**Sampling 3D Point Cloud for Gaussian Initialization.** Empirically, this strategy often produces 2D artistic fonts with clean backgrounds, facilitating the segmentation of foreground textures. Specifically, we adopt ClipSeg (Lüddecke & Ecker, 2022) $\xi$ to segment the forground heatmap as

$$\mathcal{H}_g = \xi(x_g), \tag{7}$$

under which we can obtain the font foreground with simple pre-processing (such as thresholding). Subsequently, we uniformly sample foreground pixels on the 2D mask and replicate these samples along the depth axis at fixed intervals. Stacking these parallel 2D slices forms a coarse 3D volumetric point cloud, which serves only as an initialization for 3DGS optimization, as shown in Fig. 1. Such a process does not use any depth maps or camera parameters.

### 3.3 3D Gaussians Optimization via Score Distillation Sampling

**3D Gaussian Splatting (3DGS).** 3DGS represents an 3D font by a set of 3D Gaussians as

$$\mathcal{G} = \{(\mu_i, \Sigma_i, c_i, \alpha_i)\}_{i=1}^N, \tag{8}$$

where $\mu_i$ and $\Sigma_i$ denote the mean and covariance in 3D space, while $c_i$ and $\alpha_i$ represent color and opacity. After projecting each Gaussian onto the image plane, the color of a pixel is rendered as

$$C(x) = \sum_{i=1}^N c_i \alpha_i \mathcal{N}(x|\tilde{u}_i, \tilde{\Sigma}_i) T_i, \tag{9}$$

where $\tilde{u}_i$ and $\tilde{\Sigma}_i$ are the projected mean and variance, $T_i = \prod_{j<i}(1 - \alpha_i)$ denotes the accumulated transmittance for alpha blending, and $\mathcal{N}$ is the Gaussian kernel.

**Score Distillation Sampling (SDS).** To further leverage the priors of the pre-trained 2D diffusion model $\phi_x$ for 3D font generation, we accumulatively optimize 3D Guassians $\mathcal{G}$ through SDS , which projects the differentiable 3D representation from various viewpoints and makes the projected 2D images match the text conditions. Let the differentiable 3D Gaussians $\mathcal{G}$ transform parameters $\theta$ to render 2D images as $x = \mathcal{G}(\theta)$, the optimizing gradient is computed as

$$\nabla_\theta \mathcal{L}_{SDS}(\phi, x = \mathcal{G}(\theta)) \triangleq \mathbb{E}_{t,\epsilon}\left[w(t)(\hat{\epsilon}_\phi(z^t; y, t) - \epsilon)\right], \tag{10}$$

where $w(t)$ is a weight function, $\hat{\epsilon}_\phi(z^t; y, t)$ predicts the sampled noise $\hat{\epsilon}_\phi$ conditioned on the noisy latent $z^t$ and the given text prompts $y$. By lifting 2D models into 3D, such an approach eliminates the need for real 3D data acquisition when optimizing 3D Gaussians.

### 3.4 Structure-Aware Synthesis with Part-Level Style Control

By leveraging the strong geometry priors of 3D Gaussians, we can achieve structure-aware 3D-AFG with precise part-level style control. Specifically,

**Component-Wise Style Specification**  To enable part-level style control, we decompose the printed glyph $x_p$ into $M$ glyph components, where each component $g_m$ is paired with the part-level style description $y_m$. Therefore, we obtain the part-level glyph-style annotations $\{(g_m, y_m)\}_{i=1}^M$. Therefore, we feed each component into the Glyph2Cloud to obtain the initial Gaussians $\{\mathcal{G}_m\}_{m=1}^M$, and we can also obtain the global font Gaussians $\mathcal{G}_0$ by composing components in the spatial space. Therefore, we can achieve part-level style control through component-wise SDS as

$$\nabla_\theta \mathcal{L}_{SDS} \quad = \quad \sum_{m=0}^M \lambda_i \nabla_\theta \mathcal{L}_{SDS}\left(\phi, \mathcal{G}_i(\theta)\right) \tag{11}$$

$$\triangleq \quad \mathbb{E}_{t,\epsilon,m}\left[\lambda_i w(t)(\hat{\epsilon}_\phi(z_m^t; y_i, t) - \epsilon)\right], \tag{12}$$

where $\lambda_m$ controls the importance of $m$-th part, and the pre-trained $\phi$ are shared for all Gaussians.

**Dynamic Component Assignment (DCA)**  Due to the dynamic optimization mechanism of 3DGS, Gaussian points may drift over iterations. As a result, points from different components can overlap and interfere with each other, ultimately degrading the overall generation quality. To address such drift-induced component entanglement, we propose a dynamic component assignment strategy. Specifically, we leverage the 2D stylized font and front-view projection of 3D-GS to obtain a component label map $\mathcal{M}$, where each pixel at 2D position $p$ is assigned a component label as

$$\mathcal{M}(p) = \arg\max_m \left(\log(\mathcal{H}_g^m(p) + \delta) - \beta(||p - u_\mathcal{H}^m||_2)\right), \tag{13}$$

where $\mathcal{H}_g^m$ is the 2D heatmap of the $m$-th component (according to Eq. 7), $u_\mathcal{H}^m = \frac{\sum_{q \in g_m} q\mathcal{H}_g^m(q)}{\sum_{q \in g_m} \mathcal{H}_g^m(q)}$ is the centroid position of $m$-th component, and $\delta$ is an infinitesimal. Let $\tilde{u}_i$ be the projected 2D mean of the $i$-th Gaussian point, then we dynamically update the component label of each Gaussian point as $\mathcal{M}(\tilde{u}_i)$ and re-group the Gaussian points from each component $\mathcal{G}_m$ during 3DGS optimization. This eventually helps achieve more effective and efficient part-level style control for 3D-AFG.

## 4 EXPERIMENTS

### 4.1 EXPERIMENTAL SETUPS

**Data Preparation.**  We constructed a collection of glyph–text pairs, including 10 printed digits, 26 uppercase English letters, and 8 Chinese characters. Style text prompts cover categories such as fruits, foods, and other general objects. For global style generation, printed glyphs are created from font library files. For part-level style control, the glyph is further divided into 2–3 components, each labeled with a style description. Other methods that only accept unified prompts can generate corresponding text prompts using GPT-4. Our data collection consists of 44 characters, each combined with 2 font styles and 2 modes (local or global), resulting in 1760 glyph–text pairs. Notably, we did not create or collect any realistic 3D fonts, since our SplatFont3D requires no realistic 3D data. More details of text prompt preparation refer to Appendix A.1.

**Implementation Details.**  Glyph2Cloud generated 2D images at 768×768 resolution, and the 3D fonts were rendered at 1024×1024, where Stable-Diffusion (Rombach et al., 2022) is adopted as the 2D prior model. The model was optimized using Adam with a learning rate of 0.001 and the DDPM scheduler. We set $\lambda_0 = 0.01$ for global SDS and each local $\lambda_i$ proportional to the region's area relative to the full glyph. Training and 3D rendering were performed on RTX-3090 with PyTorch.

**Evaluation Metrics.**  The following metrics are utilized to thoroughly evaluate different models:

- *Semantic Consistency:* The **CLIP** score (Radford et al., 2021; Hessel et al., 2021) and **Alignment** (He et al., 2023) assessment measure the text-style consistency of the generated 3D fonts. They quantify the correlation between the text prompt and each 2D image rendered from different views.
- *Visual Quality and View Consistency:* The **Quality** (He et al., 2023; Xu et al., 2023), **V-LPIPS** (Zhang et al., 2018), and **V-CLIP** (Radford et al., 2021) measure the visual quality and view consistency of the generated 3D fonts. They quantify the correlation between the 2D images rendered from different views, where such view consistency reflects the visual quality of 3D objects.

More details of evaluation metrics refer Appendix A.2.

**Competitors.** As 3D-AFG remains underexplored, we only compare classic text-to-3D models that can be adopted to this task, including **DreamFont3D** (Li et al., 2024), **DreamFusion** (Poole et al., 2023), **Latent-NeRF** (Metzer et al., 2023), **MVDream** (Shi et al., 2023), **Wonder3D** (Long et al., 2024), **Fantasia3D** (Chen et al., 2023a), **GSGEN** (Chen et al., 2024b), **GaussianDreamer** (Yi et al., 2024a), **GaussianDreamerPro** (Yi et al., 2024b), **Trellis** (Xiang et al., 2024).

**Synthesis Tasks.** We thoroughly evaluated different models under the following scenarios:

- *Global Style Generation.* The model produces each 3D font with a consistent global style. Input is either the glyph paired with a single style description or a corresponding unified text prompt.
- *Part-Level Style Control.* The model generates 3D fonts with distinct styles applied to different components of each glyph. Inputs consist of either multiple component images, each with a single-style description, or a unified text prompt specifying the character with part-level styles.

## 4.2 COMPARISON WITH SOTA METHODS

To demonstrate the effectiveness of our method, we compared our SplatFont3D with existing text-to-3D models through both quantitative and qualitative analyses regarding the generation performance.

| Method | Global Style Generation | | | | |
|---|---|---|---|---|---|
| | CLIP↑ | Alignment↑ | Quality↑ | V-LPIPS↓ | V-CLIP↑ |
| Wonder3D | 0.64 | 3.09 | 25.28 | 0.51 | 0.74 |
| MVDream | 0.70 | 2.81 | 29.77 | 0.36 | 0.89 |
| Latent-NeRF | 0.64 | 2.34 | 17.12 | 0.19 | 0.92 |
| GsGen | 0.66 | 3.57 | 37.17 | 0.31 | 0.92 |
| DreamFusion | 0.60 | 3.60 | 17.61 | **0.16** | 0.91 |
| GaussianDreamer | 0.71 | 3.62 | 40.36 | 0.19 | 0.92 |
| GaussianDreamerPro | 0.76 | 2.91 | 40.90 | 0.35 | 0.85 |
| Fantasia3D | 0.63 | 3.24 | 36.58 | 0.36 | 0.91 |
| Trellis | 0.72 | 2.39 | 29.01 | 0.21 | 0.91 |
| DreamFont3D | **0.82** | **4.38** | 35.62 | 0.19 | **0.96** |
| SplatFont3D (Ours) | 0.80 | 4.02 | **53.11** | 0.18 | 0.93 |
| | Part-Level Style Control | | | | |
| Wonder3D | 0.65 | 3.59 | 22.87 | 0.55 | 0.75 |
| MVDream | 0.65 | 2.81 | 22.10 | 0.26 | 0.91 |
| Latent-NeRF | 0.56 | 3.17 | 15.50 | 0.20 | 0.93 |
| GsGen | 0.68 | 3.74 | 35.21 | 0.34 | 0.92 |
| DreamFusion | 0.62 | 2.57 | 20.37 | 0.21 | 0.92 |
| GaussianDreamer | 0.70 | 3.70 | 33.74 | 0.21 | 0.93 |
| GaussianDreamerPro | 0.79 | 2.42 | 34.34 | 0.31 | 0.89 |
| Fantasia3D | 0.65 | 3.75 | 32.10 | 0.36 | 0.91 |
| Trellis | 0.74 | 2.60 | 28.91 | 0.22 | 0.88 |
| DreamFont3D | 0.81 | 3.22 | 33.82 | 0.21 | **0.95** |
| SplatFont3D (Ours) | **0.84** | **4.14** | **48.89** | 0.19 | 0.92 |
| | Global Generation + Part-Level Control | | | | |
| Wonder3D | 0.65 | 3.34 | 24.08 | 0.53 | 0.74 |
| MVDream | 0.66 | 2.81 | 25.89 | 0.31 | 0.90 |
| Latent-NeRF | 0.66 | 2.81 | 25.89 | 0.31 | 0.90 |
| GsGen | 0.67 | 3.65 | 36.19 | 0.32 | 0.92 |
| DreamFusion | 0.62 | 3.09 | 18.99 | 0.19 | 0.91 |
| GaussianDreamer | 0.71 | 3.66 | 37.05 | 0.20 | 0.92 |
| GaussianDreamerPro | 0.77 | 2.67 | 37.62 | 0.33 | 0.87 |
| Fantasia3D | 0.64 | 3.50 | 34.34 | 0.36 | 0.91 |
| Trellis | 0.73 | 2.49 | 28.95 | 0.21 | 0.89 |
| DreamFont3D | 0.81 | 3.80 | 34.72 | 0.20 | **0.96** |
| SplatFont3D (Ours) | **0.82** | **4.08** | **51.00** | **0.18** | 0.93 |

Table 1: Quantitative comparisons of different methods for 3D-AFG under different settings.

**Quantitative Results** Table 1 reports the quantitative results of different methods for 3D-AFG under different settings, including "Global Style Generation", "Part-Level-Style Control", and the

combination of the two. We can observe that our SplatFont3D achieves competing performance with existing 3D models for global style generation, while largely outperforming previous 3D models for part-level style control, especially in terms of style-text consistency and visual quality. Overall, quantitative comparisons demonstrated that our SplatFont3D achieves the SoTA performance for 3D-AFG, especially for structure-aware generation with part-level style control.

**Qualitative Results** Fig 2 illustrates the qualitative comparison of different models for 3D-AFG with "Global Style Generation" and "Part-Level Style Control". We can observe that it is inapplicable to directly adopt the existing 3D models for 3D-AFG, due to the unique challenges of 3D-AFG over 3D general object synthesis. Although DreamFont3D can generate recognizable 3D fonts of digits and letters, it struggled to synthesize 3D artistic fonts of complex structures (i.e., Chinese characters). Instead, our SplatFont3D exhibits more accurate font effects with more precise locations and better recognizability, even for Chinese characters of complex structures.

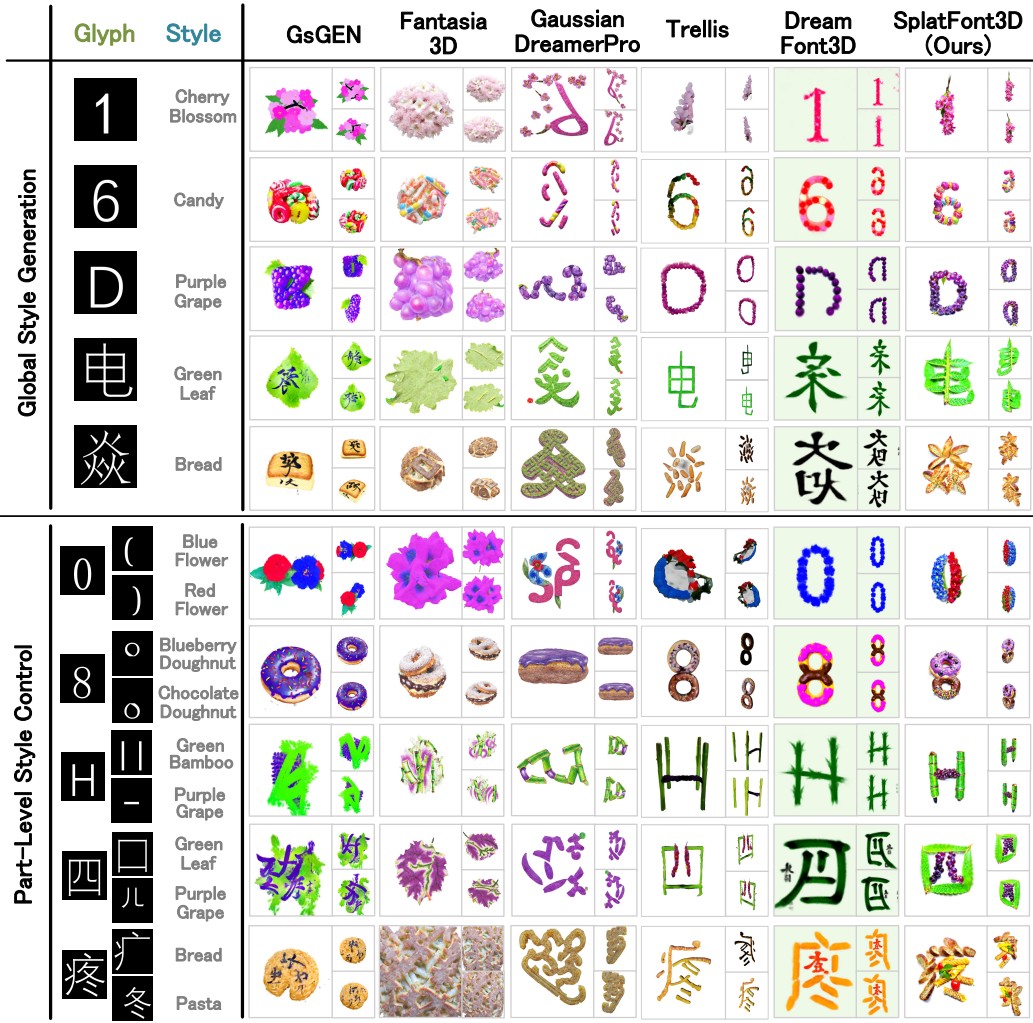

Figure 2: Qualitative comparisons of global style generation and part-level style control.

**Rendering Efficiency Comparison** Fig. 3 reports the rendering efficiency comparison of different methods for 3D-AFG on a RTX-3090 GPU. It can be observed that our SplatFont3D achieves a clear advantage in rendering speed over other 3D models. Beyond the inherent efficiency of 3DGS, this improvement is attributed to our two key designs: (1) Glyph2Cloud, which provides the well-initialized Gaussians for optimization from a better starting point, and (2) Dynamic Component Assignment, which prevents Gaussian point drifting during the optimization process. Together, these enable SplatFont3D to achieve faster and more stable rendering.

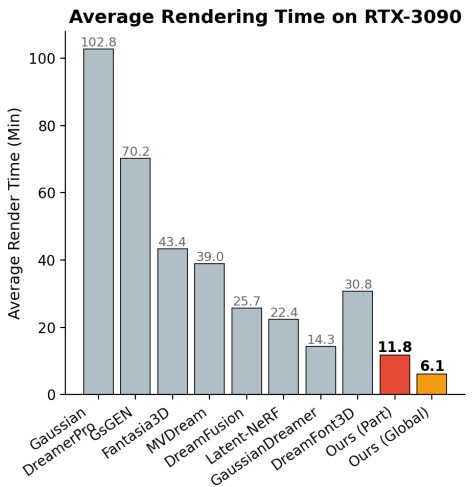

Figure 3: Rendering time comparison.

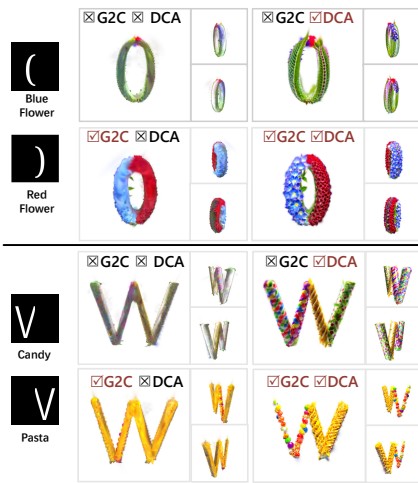

Figure 4: Qualitative ablation results.

## 4.3 ABLATION STUDY

**Quantitative and Quantitative Ablations.** To demonstrate the effectiveness of G2C and DCA, we presented quantitative and qualitative comparisons in Table 2 and Fig. 4. We can observe that: (1)Without DCA & G2C, the 3D rendering fails; (2) Without G2C, the stylization fails; (3) Without DCA, 3D fonts blur, and part-level control fails; and (4) Only with DCA & G2C, the 3D rendering succeeds. Results demonstrated their significance for 3D-AFG with precise part-level control.

| ID | G2C | DCA | Part-Level Style Control | | | | |
|----|-----|-----|--------|-----------|---------|----------|---------|
| | | | CLIP↑ | Alignment↑ | Quality↑ | V-LPIPS↓ | V-CLIP↑ |
| 1 | × | × | 0.73 | 3.42 | 36.85 | 0.26 | 0.87 |
| 2 | ✓ | × | 0.71 | 3.05 | 43.50 | 0.27 | 0.89 |
| 3 | × | ✓ | 0.77 | 3.30 | 41.38 | 0.25 | 0.86 |
| 4 | ✓ | ✓ | **0.83** | **3.94** | **47.36** | **0.21** | **0.92** |

Table 2: Ablation results on framework components.

**Comparison with 3D Methods under Same Initialization.** Most 3D baselines are text-to-3D models that receive only texts as inputs, thus making our G2C inapplicable. Therefore, we evaluated only the subset of 3D models that rely on initial point clouds or images by conducting the same initialization (e.g., either using G2C or not). As shown in Table 3, our G2C module improves the performance of most 3D models for 3D-AFG, and our SplatFont3D achieves the best results.

| Method | G2C | CLIP↑ | Alignment↑ | Quality↑ | V-LPIPS↓ | V-CLIP↑ |
|--------|-----|-------|-----------|----------|----------|---------|
| Wonder3D | × | 0.64 | 3.09 | 25.28 | 0.51 | 0.74 |
| | ✓ | 0.78 | 3.94 | 32.48 | 0.38 | 0.80 |
| Trellis | × | 0.61 | 2.85 | 20.64 | 0.20 | 0.91 |
| | ✓ | 0.72 | 2.39 | 29.01 | 0.21 | 0.91 |
| GaussianDreamer | × | 0.71 | 3.62 | 40.36 | 0.19 | 0.92 |
| | ✓ | 0.70 | 3.17 | 35.15 | 0.17 | 0.90 |
| SplatFont3D (Ours) | × | 0.72 | 3.68 | 32.79 | 0.24 | 0.87 |
| | ✓ | 0.80 | 4.02 | 53.11 | 0.18 | 0.93 |

Table 3: Quantitative comparson of different methods under the same initialization.

**Glyph2Cloud for Shape-Style Tradeoffs.** As shown in Fig. 5, we demonstrated that our G2C can achieve customized shape-style tradeoffs for 3D-AFG. By dynamically adjusting the hyperparameters $K$ and $\alpha$ in Eq. (4), it enables controllable shape-style tradeoffs in 2D and 3D.

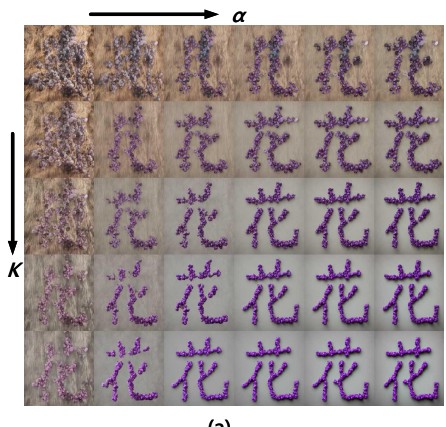
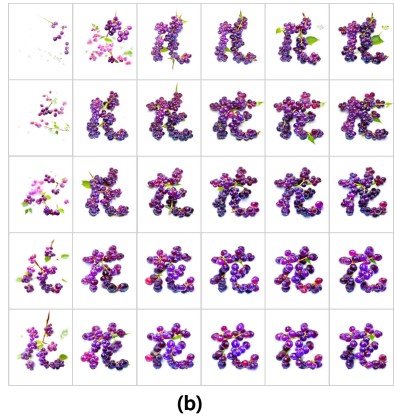

(a)

(b)

Figure 5: Glyph2Cloud for shape-style tradeoffs: (a) 2D results and (b) the final 3D fonts.

**Stroke-Level Control With Various Component Densities.** To demonstrate structure-aware generation, we further adopted our SplatFont3D for finer-grained stroke-level control. As shown in Table 4, we observe that higher component density results in higher computational costs and inferior visual quality due to increased optimization complexity. The qualitative results in Fig. 4 also demonstrate that our method can be feasibly extended to finer-grained stroke-level control.

| # of Stroke | Visual Quality | | | | | GPU (GB) | Speed (Min) |
|---|---|---|---|---|---|---|---|
| | CLIP↑ | Alignment↑ | Quality↑ | V-LPIPS↓ | V-CLIP↑ | | |
| 3 | 0.84 | 3.05 | 26.73 | 0.29 | 0.88 | 14.48 | 13.99 |
| 4 | 0.81 | 2.50 | 31.89 | 0.31 | 0.87 | 16.14 | 17.79 |
| 5 | 0.77 | 3.68 | 30.52 | 0.33 | 0.85 | 17.64 | 21.42 |
| 6 | 0.80 | 2.69 | 34.85 | 0.34 | 0.84 | 19.24 | 25.06 |

Table 4: Quantitative results on stroke-level control of various component densities.

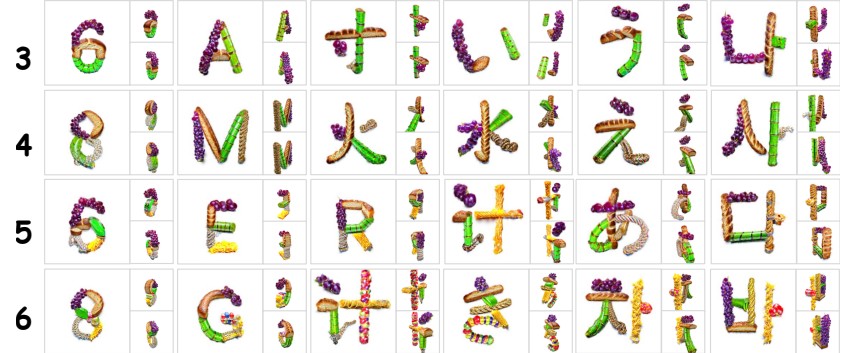

Figure 6: Qualitative results of stroke-level control with various component densities.

## 5 CONCLUSION

Most existing studies are limited to generating 2D artistic fonts, leaving the 3D artistic font generation largely underexplored. In this paper, we presented SplatFont3D, a structure-aware text-to-3D artistic font generation framework that enables fine-grained part-level style control. Specifically, our Glyph2Cloud module progressively refines 2D glyphs while preserving semantic consistency, producing well-initialized 3D point clouds for Gaussian-based modeling. By integrating 2D diffusion priors with 3D Gaussian geometry and employing a dynamic component assignment strategy, SplatFont3D effectively resolves drift-induced component entanglement, achieving explicit and controllable part-level styling without requiring real 3D font data. Extensive experiments show that our method outperforms existing text-to-3D approaches in style–text consistency, visual quality, and rendering efficiency, demonstrating its effectiveness and potential for immersive 3D applications.

## ETHICS STATEMENT

This work introduces a method for structure-aware 3D artistic font generation with part-level style control, intended for applications in digital design, creativity, and accessibility. All pretrained models used are publicly available, and all data collections are synthetically generated, and no personal or sensitive information is involved. While the technique could potentially be misused to imitate proprietary fonts, our contribution is intended solely for research and creative purposes, and we encourage responsible use. We acknowledge limitations in stylistic diversity and the environmental cost of training, and we have aimed to minimize computational overhead where possible.

## REPRODUCIBILITY STATEMENT

We have taken steps to ensure the reproducibility of our work. The full model architecture, training procedure, and hyperparameters are described in Section 3. Preparations of text prompts for different text-to-3D models are given in Appendix A.1. All evaluation metrics are formally defined in Section 4 and Appendix A.2, and all used pre-trained 2D diffusion models are publicly available. We only use the synthetically generated data collections, where the data generation processes are well described in Section 4.1. We also disclose the use of large language models (LLMs) in the Appendix, including what LLMs are used for metric evaluation and prompt construction in experiments. All used LLMs are publicly available. Moreover, comprehensive experimental details, including training configurations, evaluation scenarios, and evaluation metrics, are provided in Section 4.

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

## A  APPENDIX

### THE USE OF LARGE LANGUAGE MODELS (LLMS)

We disclose that large language models (LLMs) were used in three limited contexts:

1. **Language Polishing** – We solely used LLMs to polish the writing, specifically for spelling and grammar checking.

2. **Evaluation for Alignment Assessment** – When computing the *Alignment Assessment* (see Appendix A.2), we employed BLIP-2 to generate captions for images from each viewpoint, and then used GPT-4 to evaluate the consistency between the generated captions and the corresponding 2D view images.

3. **Prompt Construction** – for text-to-3D models that only accept text prompts as input, we used GPT-4 to generate text prompts depending on the given text–glyph pairs.

Therefore, we confirm that LLMs did not contribute to the research ideation, methodology, experimental design, analysis, or substantive writing of this paper.

### A.1  PREPARATION OF TEXT PROMPTS FOR DIFFERENT MODELS

Since there is no available dataset that provides text descriptions for 3D artistic fonts, we must construct all text prompts ourselves. Specifically, we prepare text prompts as follows:

- **Simple style descriptions for our SPlatFont3D**: For our own Glyph2Cloud pipeline, we use positive text prompts such as "a professional photograph of some objects on a black table"; and for 3D optimization part, the positive text prompt is "{*object*} *style,* {*front|side|overhead*} *view.*" similar to DreamFusion. We find that these simple text prompts already produce strong results for our method, and using more complex descriptions offers no significant improvement.

- **GPT-generated prompts for other 3D models** For other text-to-3D models that require richer text prompts, we use GPT-4 to automatically generate text prompts depending on the given style keywords. For example, given text keywords that identify styles, GPT-4 produces a detailed text prompt for these 3D methods. Here is a GPT-generated example: "*An English letter 'C', with the upper half in green bamboo style and the lower half in rope style.*"

## A.2 DETAILS OF EVALUATION METRICS

- **CLIP:** Measures the semantic fidelity of generated glyphs by encoding multiple rendered views with CLIP (Radford et al., 2021; Hessel et al., 2021) and computing the cosine similarity to the corresponding textual prompt, then averaging across views to obtain a robust multi-view score.
- **Alignment Assessment:** Evaluates higher-level semantic correspondence by generating captions for each view with BLIP-2 (Li et al., 2023a), consolidating them into a single summary using GPT-4, and then prompting the model to rate the alignment between the summary and the original prompt on a five-point scale.
- **Quality Assessment:** Evaluates the visual fidelity of generated glyphs by applying ImageReward (Xu et al., 2023) to multi-view renderings conditioned on the input prompt. To reduce view-to-view noise, each score is smoothed using a local neighborhood average over adjacent views, producing a more consistent assessment of overall image quality.
- **V-LPIPS:** Measures multi-view perceptual consistency of generated artistic glyphs by computing LPIPS (Zhang et al., 2018) between adjacent rendered views and averaging the results. This metric captures how smoothly the glyph's appearance transitions across viewpoints, reflecting both structural and stylistic coherence.
- **V-CLIP:** Evaluates multi-view semantic consistency of generated artistic glyphs by computing the cosine similarity between CLIP embeddings of adjacent views and averaging the results, capturing how consistently the glyph preserves the intended semantics across viewpoints.

## A.3 MORE QUALITATIVE RESULTS OF OUR SPLATFONT3D

As shown in Fig. 7, we also provide more qualitative results of our SplatFont3D for structure-aware 3D-AFG with part-level style control. Experimental results demonstrate the effectiveness of our approach in achieving both structural fidelity and flexible style manipulation for structure-aware 3D-AFG with customized part-level style control.

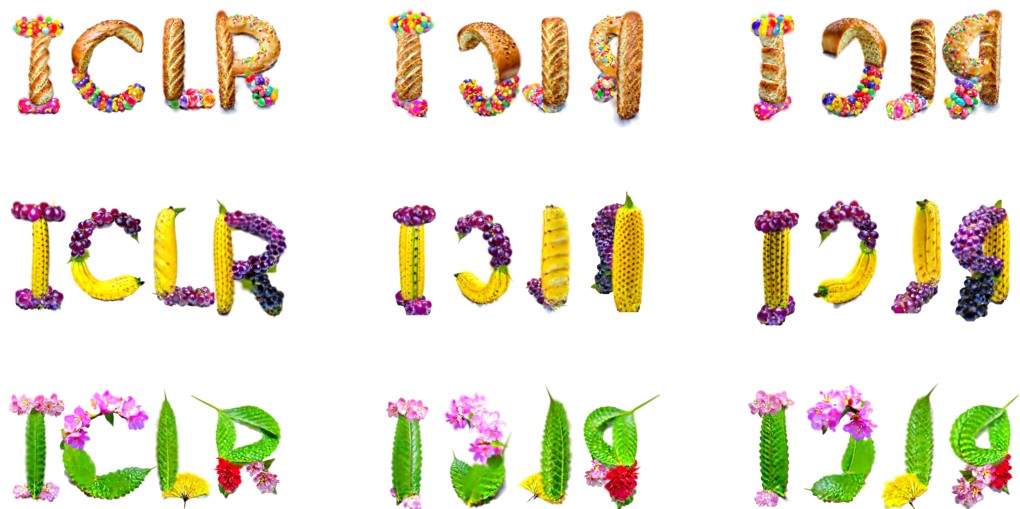

Figure 7: Qualitative results of our splatFont3D for structure-aware 3D-AFG.

## A.4 RESULTS FOR 3D-AFG WITH MORE STYLES AND LANGUAGES

We extend our evaluation data with eight new visual styles and two more languages (including Korean and Japanese), where the extended data covers five languages, 16 different styles for global, and $C_{16}^2$=120 combinations for two-part. It is worth noting that our evaluation of all methods on original data requires nearly 50 GPU-days. Due to the trade-off between cost and feasibility, we only compare SplatFont3D with several SOTA methods as shown in Table 5. The results show that our method generalizes well across these new settings and outperforms previous SOTA. The qualitative

results in Fig. 8 also demonstrate the generalization ability for 3D-AFG with more diverse styles and languages.

| Method | CLIP↑ | Alignment↑ | Quality↑ | V-LPIPS↓ | V-CLIP↑ |
|---|---|---|---|---|---|
| Trellis | 0.61 | 2.53 | 18.92 | 0.22 | 0.90 |
| GaussianDreamer | 0.69 | 2.97 | 35.61 | **0.20** | 0.89 |
| DreamerFont3D | 0.77 | 3.26 | 33.19 | 0.23 | **0.93** |
| SplatFont3D (Ours) | **0.79** | **3.70** | **46.20** | 0.21 | 0.92 |

Table 5: Comparison of different methods for global style generation on extended evaluation data.

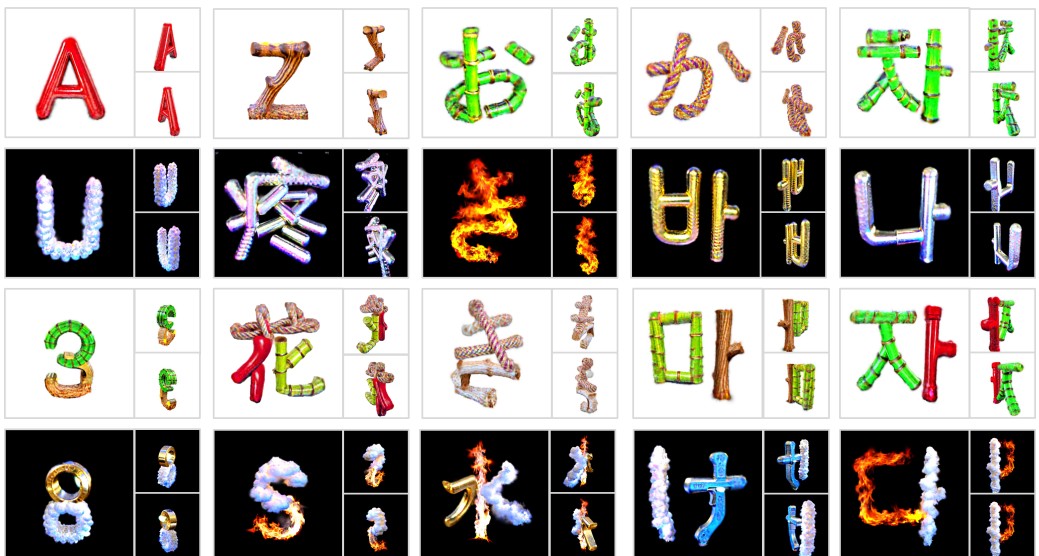

Figure 8: Qualitative results with more styles and languages. Because certain styles (such as "Cloud") feature a predominantly white foreground, we adopt a black background in the renderings to preserve clear visual contrast.

## A.5 COMPARISON WITH 3D-EDITING FOR STRUCTURE-AWARE 3D-AFG

In our SplatFont3D, 3DGS optimization and part-level control are performed synchronously, with localized control serving as a form of regularization during 3D generation. Instead, 3D editing approaches perform 3D generation and localized control sequentially, i.e., they first generate 3D assets and then apply local 3D edits. Although 3D editing with local controls is a feasible approach for 3D-AFG, we demonstrate that one-pass 3D generation is easier and more efficient than 3D editing with two-pass 3D optimization.

| # of Stroke | Method | Visual Quality | | | | | Speed↓ |
|---|---|---|---|---|---|---|---|
| | | CLIP↑ | Alignment↑ | Quality↑ | V-LPIPS↓ | V-CLIP↑ | (Min/R.) |
| 3 | GaussianEditor | 0.71 | 2.87 | 12.89 | 0.29 | 0.87 | 29.16 |
| | TIP-Editor | 0.73 | 2.63 | 15.91 | 0.31 | 0.88 | 37.37 |
| | SplatFont3D (Edit) | 0.76 | 2.94 | 19.72 | 0.35 | 0.86 | 20.23 |
| | SplatFont3D (Gen) | **0.84** | **3.05** | **26.73** | **0.29** | **0.88** | **13.99** |
| 4 | GaussianEditor | 0.63 | 1.87 | 12.40 | 0.32 | 0.86 | 38.51 |
| | TIP-Editor | 0.69 | 1.65 | 19.09 | 0.33 | 0.86 | 60.38 |
| | SplatFont3D (Edit) | 0.70 | 1.89 | 23.14 | 0.37 | 0.85 | 24.09 |
| | SplatFont3D (Gen) | **0.81** | **2.50** | **31.89** | **0.31** | **0.87** | **17.79** |
| 5 | GaussianEditor | 0.63 | 3.04 | 19.74 | 0.34 | 0.85 | 45.99 |
| | TIP-Editor | 0.67 | 3.12 | 21.59 | 0.34 | 0.85 | 88.56 |
| | SplatFont3D (Edit) | 0.63 | 2.95 | 23.38 | 0.39 | 0.85 | 27.77 |
| | SplatFont3D (Gen) | **0.77** | **3.68** | **30.52** | **0.33** | **0.85** | **21.42** |

Table 6: Quantitative results on stroke-level control of various component densities.

As shown in Table 6, we provided a quantitative comparison with 3D editing models, where the initial 3D fonts for 3D editing are generated with our SplatFont3D using global generation. In the table, "SplatFont3D (Gen)" refers to directly generating from scratch, and "SplatFont3D (Edit)" refers to firstly generating global 3D assets and then performing 3D editing with local edits.

It can be observed that 3D editing with two-pass 3D optimization performs inferior to 3D generation from scratch with one-pass 3D optimization, such as lower visual quality, less I/O overhead, and slower rendering speed. This is because the 3D editing requires redundant two-pass 3D optimization and must reconcile local edits with an already-fixed geometry, which often leads to artifacts and slower rendering efficiency. Moreover, most 3D editing methods (such as GaussianEditor and TIP-Editor) can edit only one style or one component per invocation, and thus editing multiple components requires multiple rounds of 3D optimization, which leads to accumulated errors and increased rendering time. In contrast, our one-pass generation integrates global and local objectives jointly from scratch, enabling more consistent optimization and more efficient computation. Qualitative comparison in Fig. 9 also demonstrates that direct 3D generation is more feasible than 3D editing for 3D-AFG, especially for finer-grained controls on multiple components.

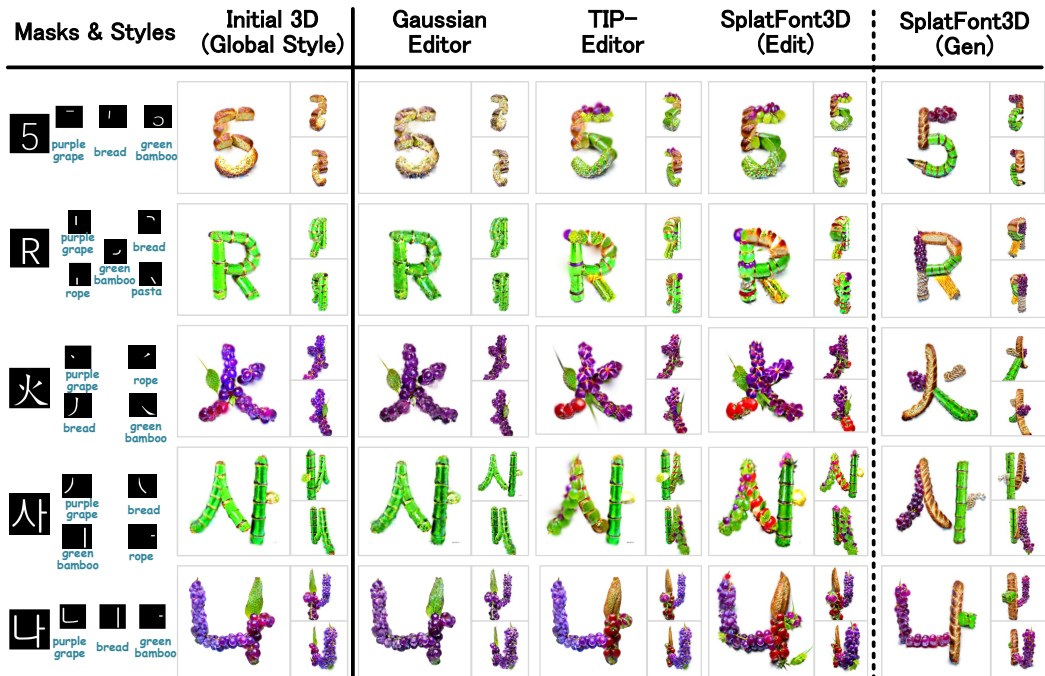

Figure 9: Qualitative comparison of different methods for part-level style control.

### A.6 MORE QUALITATIVE COMPARISONS OF DIFFERENT MODELS

To further demonstrate the effectiveness of our method for 3D-AFG, we provide a more thorough qualitative comparison between our SplatFont3D and existing text-to-3D models regarding the generation performance, including the **Global Style Generation in Fig. 10** and **Part-Level Style Control in Fig. 11**. These comparisons show that our SplatFont3D produces more faithful global styles and provides finer part-level control than existing approaches, achieving more consistent global styles and finer-grained part-level control for 3D-AFG.

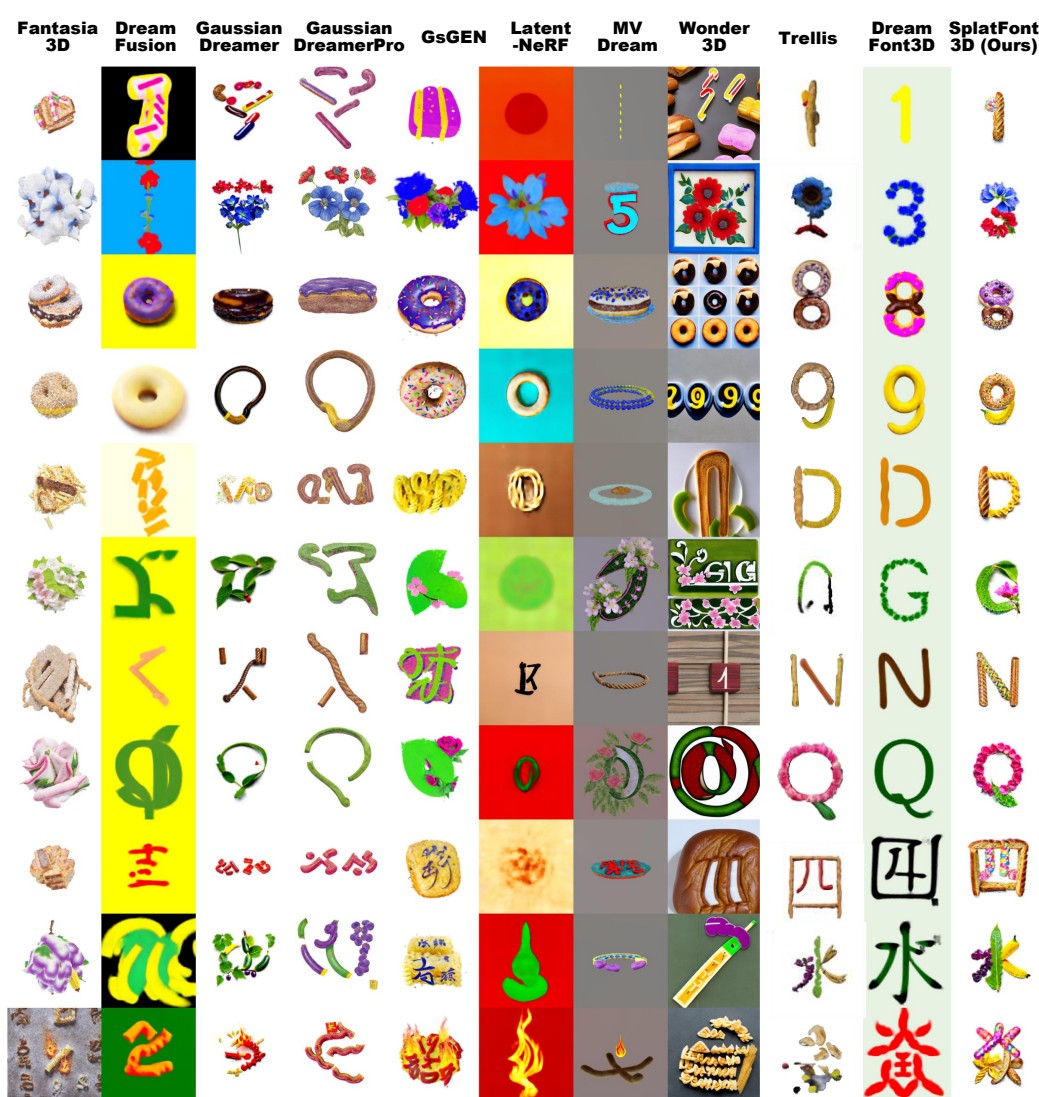

Figure 10: Qualitative comparison of different methods for part-level style control.

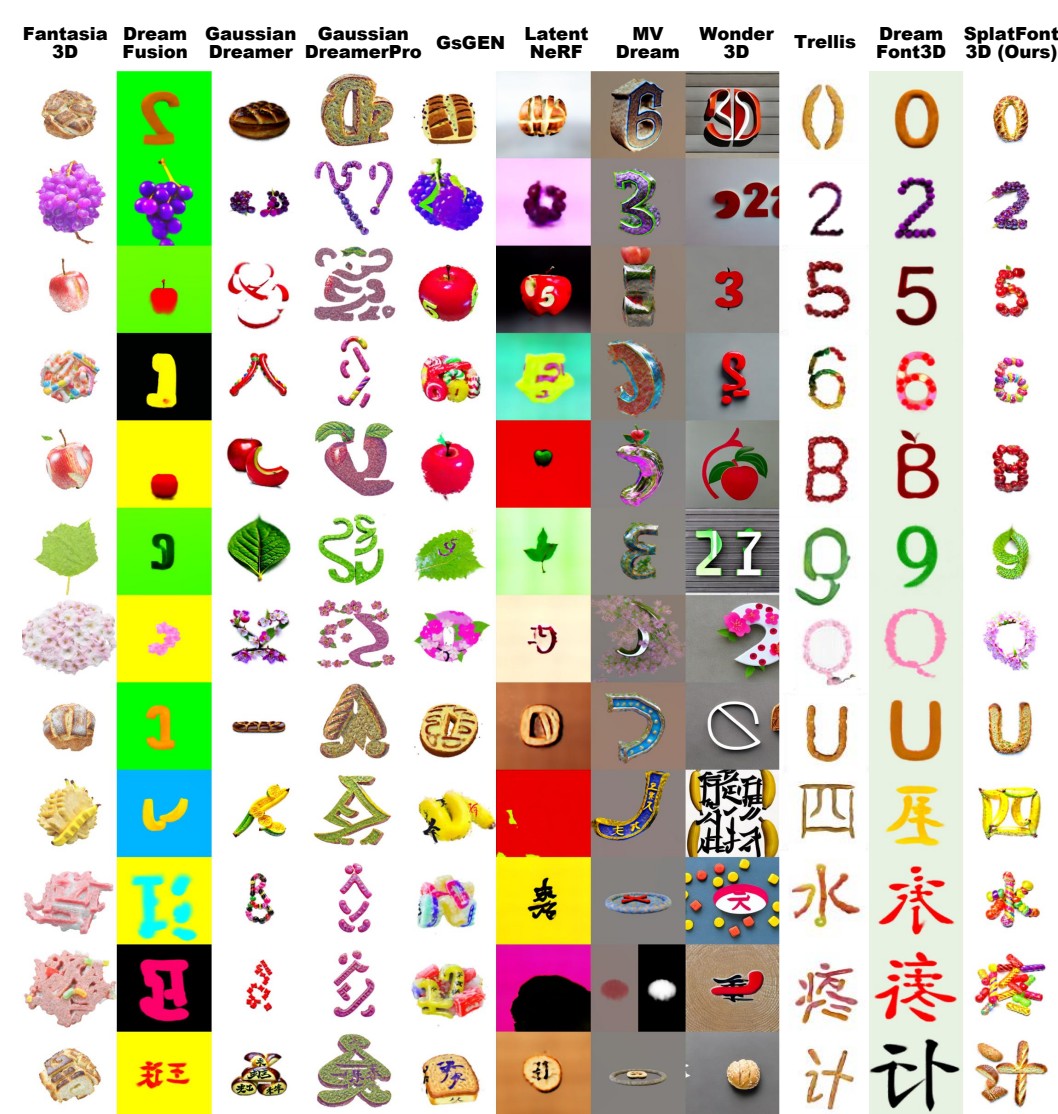

Figure 11: Qualitative comparison of different methods for global style generation.

