# OpenReview forum: "SplatFont3D: Structure-Aware Text-to-3D Artistic Font Generation with Part-Level Style Control"
_ICLR.cc/2026/Conference — Submitted to ICLR 2026_

### Official Review · Reviewer_cDMy · 2025-10-29

**Soundness:** 2
**Presentation:** 3
**Contribution:** 2
**Rating:** 2
**Confidence:** 4

**Summary:**

This paper introduces SplatFont3D, a model for structure-aware text-to-3D artistic font generation with precise component-level style control. The method combines 3D Gaussian Splatting with a 2D diffusion prior-driven Score Distillation Sampling (SDS). It includes a Glyph2Cloud module for geometry-aware 3D Gaussian initialization from 2D glyphs and introduces a dynamic component assignment strategy to address part disentanglement issues during optimization.

**Strengths:**

- The paper clearly articulates the need for and importance of 3D artistic fonts with fine-grained style control, as well as the limitations of previous NeRF/3DGS-based models in handling font semantic constraints.

- It proposes Glyph2Cloud, a method for initializing global 3D representations, and a dynamic component assignment strategy for local component editing, achieving geometric information grouping and iterative optimization for disentanglement.

- Experiments demonstrate that SplatFont3D has significant advantages in generation quality, controllability, and training efficiency.

**Weaknesses:**

- The introduction is overly verbose, with Lines 54-78 and Lines 79-96 containing repetitive content. Figure 1 lacks formulaic symbol annotations, making it difficult to understand the formulaic symbols in the method section.

- The multi-view consistency is insufficient in Table 4.2. For example, in Figure 6, the side view of the leaves still appears wide. Regarding efficiency, while using 3DGS for geometric representation is an obvious way to improve efficiency, the three-stage training strategy and component-wise independent SDS optimization further reduce efficiency.

- Technically, the paper primarily includes mask-guided 3D artistic font generation and an additional 3D editing framework. The main contribution should lie in the latter, rather than the 3DGS architecture and its advantages. Therefore:
1. The paper lacks references related to 3D editing, such as GaussianEditor, Control3D, 3DStyleGLIP, SketchDream, TIP-Editor, etc.
2. Comparative methods are missing. While the authors compare a large number of text-to-3D works, which is reasonable for evaluating 3D artistic font generation capability, they should comprehensively discuss the advantages of the proposed 3D editing method.

- Key technical details are missing. For example: How are the 2D font masks decomposed? What 2D prior model is used? How are the generated artistic font images initialized into 3D Gaussians? The authors only use frontal view constraints; how is consistency ensured for other views, such as side views?

**Questions:**

- In Section 3.2, many details are unclear. How are 3D Gaussians obtained from the generated images? What technology is used? Text-to-artistic font image generation technologies are now quite mature, such as SeeDream4.0, Hunyuan, or ControlNet. What advantages does Glyph2Cloud have over these methods?

- In the parameterized trade-off experiment for shape and style, why is there a clear gradient in the 2D generation results but not in the 3D generation?

- What are the requirements for the input text prompts? The paper does not seem to include any complete examples of prompt phrases.

- Reference the questions raised in the Weaknesses.

---

> ### Author Response · Authors · 2025-11-20
> **Author Responses to Reviewer cDMy (1/4)**
>
> ## Point-to-Point Response
>
> We sincerely thank the reviewer for encouraging the presentation of our work. Based on your valuable feedback, we have thoroughly revised the manuscript and addressed your concerns as detailed below.
>
> > ### **Q1: Technically, the paper primarily includes mask-guided 3D artistic font generation and an additional 3D editing framework. The paper lacks references related to 3D editing.**
> >
> > Thank you for the reviewer’s thoughtful comment. We would like to clarify that our method is fundamentally a **3D artistic font generation model** conditioned on 2D font mask images and style text prompts, **rather than a combination of a 3D generation model and an additional 3D editing framework**.  To avoid confusion, we will add a short subsection in **Related Work** explicitly discussing the difference between our task and 3D editing, and **cite the corresponding literature**.  More detailed clarification is as follows:
>
> **1. Difference Between 3D Editing and 3D Generation.**
>
>   - **3D generation** refers to *creating* new 3D content **from scratch** using an input such as text, an image, or some high-level specification. For example, a standard 3D-GS Splatting generation pipeline is:
>
>     > - **Step 1.** Generate an **initial and coarsen 3D point cloud** from scratch.
>     > - **Step 2.** Start from initial 3D point cloud and Optimize 3D Gaussian representation.
>
>   - While **3D editing** means *modifying* an already existing 3D asset, i.e.,modify or enhance something that already exists, rather than creating it from scratch.  The pipeline of 3D editing generation is:
>
>     > - **Step 1.** Input a **well-optimized 3D representation**.
>     > - **Step 2.**  Edit this 3D representation, and modify it into another.
>
> **2. Our pipeline follows a standard 3D-GS splatting generation pipeline**
>
> Our method is a **3D artistic font generation model**, *not a 3D editing approach*. Because,
> - **Both `Global Style Generation` and `Part-Level Style Control` of our method generate 3D fonts from scratch, rather than editing existing 3D fonts.** Specifically, our SplatFont3D  can achieve either `Global Style Generation` or `Part-Level Style Control` for 3D-AFG, and *It receives 2D font mask images (as shape constraints) and style texts as inputs, instead of existing 3D assets.*  *Thus our method belongs to 3D generation rather than 3D editing.*
> - **G2C only generate a coarse point cloud for 3D-GS initialization—rather than creating a high-quality 3D Representation.**
>   *This initial point cloud generated by G2C is only used as a starting point for the 3D-GS optimization process; it is not a fully optimized 3D representation.*
>   Consequently, our pipeline is different from 3D editing methods, which typically modify an existing high-quality 3D representation. We do **not** perform modifications from one optimized 3D representation into another.
>
> **3. Our method is consistent with existing 3D generation methods**
>
> - Existing text-to-3D models like `"GaussianDreamer" and "GaussianDreamerPro"` first adopt pretrained text-conditioned 3D diffusion models (e.g., Point-E) to generate an initial 3D point cloud, and then perform 3D-GS optimization.
>
> - Our SplatFont3D recieves`"masked font image + style text"` as inputs and employs G2C to obtain the initial 3D point cloud, and then perform 3D-GS optimization via SDS + DCA.
>
> Therefore, *our approach is fully consistent with the **3D generation** paradigm adopted by these methods, rather than a hybrid of “3D generation + 3D editing” pipeline*.
>
> >---
> >
> >> ### **Q1-1: The paper lacks references related to 3D editing, etc.**
> >
> >> ### **Q1-2: The authors should comprehensively discuss the advantages of the proposed 3D editing method.**
> >
> >**Response to Q1-1 and Q1-2:**
> >
> >- We thank the reviewer for these insightful comments. We have included a subsection in **Related Work** explicitly discussing the difference between 3D generation and 3D editing, and **have included references to relevant 3D-editing works**.
> >- As noted in our previous responses to your question **Q1**, *our method is consistent with existing 3D generation methods, and is different from 3D editing models*, and also is *not* a hybrid of “3D generation + 3D editing.” Therefore, while we do compare extensively with text-to-3D generation methods, our approach does not aim to provide additional 3D editing functionality or advantages.
>
> We sincerely hope that our clarifications will satisfactorily address the reviewer’s concerns.

---

> ### Author Response · Authors · 2025-11-20
> **Author Responses to Reviewer cDMy (2/4)**
>
> > ### **Q2-1:How are 3D Gaussians obtained from the generated images? What technology is used?**
> >
> > Thanks for your constructive suggestion, and **more details are now clarified at the beginning of Section 3.2**.   Specifically,
>
> - The core idea is to **first segment the foreground glyph from the generated image**, then **uniformly sample 2D pixels on the foreground region**, and finally **project these sampled 2D points into 3D space**. This pipeline produces an **initial and coarse 3D point cloud**, *which is essential for initializing 3D Gaussian Splatting (3D-GS), since 3D-GS requires a reasonably good coarse 3D point cloud before optimization*.
>
> - Importantly, in Section 3.2, **G2C** focuses solely on generating the **initial coarse 3D point cloud for Gaussian initialization**, *not* on producing a fully optimized 3D Gaussian representation. **The obtained point cloud simply serves as the starting point from which 3D-GS Splatting performs its subsequent optimization.**
>
> >
>
> > ### **Q2-2: What advantages does Glyph2Cloud have over these mature text-to-artistic methods (e.g., SeeDream4.0, Hunyuan, or ControlNet)?**
> >
> > Thank you for your insightful comment—this is indeed an important and very good question, **as the advantages of Glyph2Cloud over those powerful 2D models essentially form one of the central contributions of our work.** Our clarification is as follows:
>
> **1. G2C is built upon the mature text-to-image models with Denoising Intervention**
>
> *Glyph2Cloud is developed on top of mature text-to-image generation technologies (specifically **Stable Diffusion**)*, in which we introduce an extra denoising intervention to modify the sample procedure when inferencing. Crucially, *we do not modify any pretrained model weights of 2D stable-diffusion* in Fig.1; instead, we **redesign the sampling procedure through denoising intervention**  (i.e., injecting the shape latent $z_s$ into the original latent noise $z^t$ as formulated in Eq. (4)~(6)) to control the shape–style trade-off, enabling controllable 3D-aware generation in both structure and style.
>
> **2. Same Generation Capability, Different Sampling Strategy**
>
> Because Glyph2Cloud relies entirely on the priors of **Stable Diffusion** and leaves all pretrained parameters untouched, *its 2D image generation capability is inherently comparable to SeeDream4.0, Hunyuan, and ControlNet*. The G2C module can also be integrated with these off-the-shelf models (by replacing StableDiffusion), as **the key innovation lies not in the 2D prior model (e.g. Stable Diffusion, ControlNet, etc.) but in our sampling strategy and denoising intervention**.
>
> >
>
> >  ### **Q3: In the trade-off experiment for shape and style, why is there a clear gradient in the 2D generation results but not in the 3D generation?**
> >
> >  Thank you for your insightful question. We hope we can address your concerns through the following explanations:
>
> - **Clear gradient in 2D**: In the 2D generation stage, the results are directly influenced by the *denoising intervention* introduced in the G2C module, so the shape–style trade-off manifests clearly in the 2D outputs
>
> - **Weaker gradient in 3D**: In contrast, during the 3D-GS optimization, only 3D Gaussians are optimized via SDS loss, while shape–style constraint through denoising intervention is no longer feasible (since 2D projected images are rendered by 3D-GS rather than  2D Stable-Diffusion).  Its gradient on 3D heavily relies on the initialized 3D point cloud obtained from G2C initialization.
>
> - **G2C still enables meaningful 3D style-shape trade-off**: Fig. 5 shows we can still achieve a meaningful—though less pronounced—shape–style trade-off in the final 3D outputs.
>
>
> >
>
> > ### **Q4: What are the requirements for the input text prompts?**
> >
> > Thanks for your good comments. As suggested, we have now clarified this in `"Appendix A.1"` .   Specifically, we prepare text prompts as follows:
>
> - **Simple style descriptions for our method**: For our own Glyph2Cloud pipeline, we use positive text prompts such as `"a professional photograph of some {object}s on a black table"`; and for 3D optimization part, the positive text prompt is `"{object} style, {front|side|overhead} view"` similar to DreamFusion. We find that these simple text prompts already produce strong results for our method, and using more complex descriptions offers no significant improvement.
> - **GPT-generated prompts for other 3D models**: For other text-to-3D models that require richer text prompts, we use GPT-4 to automatically generate text prompts depending on the given style keywords. For example, given text keywords that identify styles, GPT-4 produces a detailed text prompt for these 3D methods. Here is a GPT-generated example: `“An English letter 'C', with the upper half in green bamboo style and the lower half in rope style.”`

---

> ### Author Response · Authors · 2025-11-20
> **Author Responses to Reviewer cDMy (3/4)**
>
> > ### **Q5: Key technical details are missing. For example:**
> >
> > ---
> >
> > #### **Q5-1: How are the 2D font masks decomposed?**
>  >
>  >  To enhance practicality, we allow users to **manually specify the components in the 2D font mask**. This enables **customized part-level control**, similar to how one would edit photos in software like `Photoshop`. For convenience, we can also consider providing decomposition scripts to help users automatically split font masks.
> >
> > ---
> >
>  > #### **Q5-2: What 2D prior model is used?**
>  >
> >  As noted in our previous responses to to your question **Q2-2**, the used 2D prior model is **Stable-Diffusion**.
> >
> > ---
> >
>  >  #### **Q5-3: How are the generated artistic font images initialized into 3D Gaussians**
>  >
> >  As noted in our previous responses to your question **Q2-1** and **the beginning of Section 3.2**, the core idea is to
>  >
>  >  - First **segment the foreground glyph from the generated image**,
> >
> >  - Then **uniformly sample 2D pixels on the foreground region**,  and
> >
> >  - Finally **project these sampled 2D points into 3D space to form initial 3D point cloud**.
>  >
> >  *Any post-processing strategy following this pipeline* is acceptable for obtaining the initial 3D point cloud.
>  >
> >  Such an initial and coarse 3D point cloud simply serves as the starting point of 3D-GS Splatting for performing its subsequent optimization. This is different from 3D editing, which requires well-optimized 3D inputs.
> >
> > ---
> >
> > #### **Q5-4: How is consistency ensured for other views, such as side views**.
> >
> >  The **Score Distillation Sampling (SDS) loss** [1] is applied to optimize the 3D Gaussians, which naturally enforces multi-view consistency.  Specifically,
>  >
> >  - We first obtain a 3D Gaussian initialization via G2C depending on the frontal view, but after that, we optimize the 3D  Gaussian representations from different views through SDS loss, which renders 2D images by projecting 3D Gaussians from different viewpoints and validates the visual qualities of those projected images using a  2D prior model (e.g., Stable-Diffusion).
>  >
>  >  - Since the SDS loss operates **on the shared 3D Gaussian parameters**, which define the full 3D structure, updates are not limited to a single 2D image. Optimizing from any view indirectly constrains other views, because all rendered images share the same underlying 3D representation. This naturally enforces consistency across views, including side and oblique angles. More details refer to Dreamfusion [1].
>  >
>  >  [1] Dreamfusion: Text-to-3d using 2d diffusion. In *ICLR*, 2023.
>
> >
>
> > ### **Q6-1: The multi-view consistency is insufficient in Table 4.2.**
> >
> > Thanks for your insightful comments. This is because our method does not rely on any 3D supervision, but it attains unique advantages over 3D supervised methods. More specifically,
>
> **1. The reason is the lack of 3D supervision, which affects consistency**
>
> Training directly on large-scale 3D data would naturally yield the strongest multi-view consistency. However, our method does not rely on any 3D supervision, which inevitably leads to certain limitations in multi-view consistency.
>
> **2. However, large-scale 3D font data is unavailable and costly**
>
> However, there is currently no large-scale 3D artistic font dataset that would allow supervised training for 3D-AFG. Constructing such a large dataset requires professional designers and is prohibitively expensive. Instead, training on small-scale datasets would cause severe overfitting and poor generalization to unseen characters and styles.
>
> **3. Advantages of our method: SplatFont3D trades consistency for strong generalization and requires no 3D real data**
>
> Although our method may exhibit weaker multi-view consistency, its key advantage is that it does not require any real 3D artistic fonts for supervision. By leveraging the strong priors of pretrained 2D generative models, our approach achieves excellent generalization and can directly synthesize 3D artistic glyphs in arbitrary styles without relying on real 3D datasets.
>
> **4. Quantitative results in Table 1 demonstrate the SoTA multi-view consistency of our method**
>
> As shown in the quantitative comparisons in Table 1, our method achieves the best overall performance for the task of 3D artistic font generation, including on multi-view consistency metrics. Nevertheless, we acknowledge that this is an emerging task that is underexplored but with substantial room for future improvements.

---

> ### Author Response · Authors · 2025-11-20
> **Author Respone to Reviewer cDMy (4/4)**
>
> > ### **Q6-2:  Regarding efficiency, the three-stage training strategy and component-wise independent SDS optimization further reduce efficiency.**
>
> Thanks for your good question, and our clarification is as follows:
>
> **1. Our method attains a faster rendering speed than other 3D models, as shown in Fig. 3**
>
> Our method is built upon 3DGS, whose inherently efficient rendering pipeline enables significantly faster rendering than other 3D models (e.g., NeRF). As shown in Fig. 3, our SplatFont3D attains a faster rendering speed than other 3D models.
>
> **2. This indeed exists a trade-off between fine-grained generation and rendering efficiency.**
>
> We agree that a finer-grained part-level generation will inevitably increase the computation cost.  We have conducted experiments to analyze how component density would impact optimization complexity and computational overhead. As shown in Table 4, we observe that higher component density results in higher computational costs and inferior visual quality due to increased optimization complexity.  We also give qualitative results in Fig. 6 in the revised manuscript.
>
>
> | # of Strokes | CLIP$\uparrow$ | Alignment$\uparrow$ | Quality$\uparrow$ | V-LPIPS$\downarrow$ | V-CLIP$\uparrow$ |GPU (GB) $\downarrow$| Speed (Min/R.)$\downarrow$|
> | :----------: | ---: | --------: | ------: | ------: | -----: | -------: | ----------: |
> |      3       | 0.84 |      3.05 |   26.73 |    0.29 |   0.88 |    14.48 |       13.99 |
> |      4       | 0.81 |      2.50 |   31.89 |    0.31 |   0.87 |    16.14 |       17.79 |
> |      5       | 0.77 |      3.68 |   30.52 |    0.33 |   0.85 |    17.64 |       21.42 |
> |      6       | 0.80 |      2.69 |   34.85 |    0.34 |   0.84 |    19.24 |       25.06 |
>
> **3. Our training strategy is TWO-stage: (1) 3D Point Cloud  Initialization via C2C (2) 3DGS Optimization via SDA & DCA**
>
> Our SplatFont3D consists of G2C and 3D-GS optimization, where *the first stage* is G2C for initialization, and *the second stage* is 3D-GS optimization. Our method can achieve either `Global Style Generation` or `Part-Level Style Control` for 3D-AFG, where the upper part of Fig.1 corresponds to 3D Global Style Generation (i.e., "G2C" + "3DGS via SDS"), and the bottom part of Fig. 1  corresponds to 3D Part-Level Control (i.e., "G2C" + "3DGS via SDS &DCA").
>
> ***(Note to the AC:  Figure 1 has been further updated after our Round-2 clarification.)***
>
> >
> >
>
>
> > ### **Q7: Other Presentation issues**
> >
> > Thanks for your constructive comments, and as suggested, we have addressed the issues in the revision. Specifically,
> >
> > ---
> >
> >   **Q7-1: The introduction is overly verbose, with Lines 54-78 and Lines 79-96 containing repetitive content.**
> >
> >  As recommended, we have addressed the verbosity by compressing this section from **Lines 79–96** to **Lines 79–88**, retaining only the essential content.
> >
> >  **Q7-2: Figure 1 lacks formulaic symbol annotations, making it difficult to understand.**
> >
> >  As suggested, **we have significantly revised Fig.1** and included more formulaic symbol annotations to make it easier to understand.

---

> > ### Comment · Reviewer_cDMy · 2025-11-26
> > **Thanks for the reply**
> >
> > Thank you for the author's reply. While the response addressed a small portion of my concerns, I still have some differing opinions and look forward to further clarification from the author.
> >
> > 1. I don't doubt that this work is 3D generation. However, the second stage of this work is closely related to 3D editing techniques. Therefore, the lack of technical and experimental discussion on 3D editing diminishes the quality of the paper. My reasons are as follows:
> >
> > 1)As shown in Figure 1, the generation target is a combination of bread and candy. However, the first stage generates 3D typography composed solely of bread, while the second stage introduces localized modifications to incorporate the candy style. Why was the candy style not integrated into the typography in the first stage, instead of being added as a localized control in the second stage?
> >
> > 2)The authors state that the first stage involves coarse-grained Gaussian points. However, the results presented in the first stage depict a refined 3D typography with a bread-style aesthetic, as shown in Figure 1. This level of detail is sufficient to incorporate other 3D editing techniques for localized control.
> >
> > II. How is the process of "projecting these sampled 2D points into 3D space" implemented? Does it require depth information and camera parameters?
> >
> > Has the motivation and principle behind "redesigning the sampling procedure through denoising intervention" been validated?
> >
> > Summary​
> > Although I maintain my opinion on technical pipline, this does not undermine the main contributions of the paper. Given the authors' improvements to the existing issues and their partial clarification of my questions, I am willing to slightly increase my score. While the paper successfully achieves a novel typography generation task, the quality of the generated results still lags significantly behind current mainstream, open-source 3D generation methods, particularly in terms of multi-view consistency.

---

> ### Author Response · Authors · 2025-11-20
> **Summary of our Revisions**
>
> ## Summary of our Revisions
>
> - As suggested, we have carefully revised Fig.1 and have included more formulaic symbol annotations to make our framework easier to understand.
> - We have cited the related literature of 3D editing and added a subsection **`3D Editing`** in related work to discuss the difference and relations between 3D-Generation and 3D-Editing. It is worth noting that:
>   - As shown in Fig.1, our SplatFont3D can achieve either `Global Style Generation` or `Part-Level Style Control` for 3D-AFG. *Our method receives 2D font mask images (as shape constraints) and style texts as inputs, instead of existing 3D assets.*
>
>   - Both `Global Style Generation` and `Part-Level Style Control`  **generate 3D fonts from scratch, rather than editing existing 3D fonts.** *Thus our method belongs to 3D generation rather  than 3D editing.*
> - As suggested, at the beginning of Section 3.2, we have given the core idea of how to obtain the initial 3D point cloud for 3D Gaussian initialization.
> - As suggested, we have given the details of text prompt preparations for different text-to-3D models in Appendix A.1.
> - As suggested, in Table 4 and Fig. 6, we have analyzed the training efficiency, like component-wise independent SDS of various component densities. Specifically, in Section 4.3, we have evaluated stroke-level style control of various component densities and have analyzed how such an extension impacts optimization complexity and computational overhead.
> - As recommended, we have carefully revised the verbose part in the Introduction section (i.e., Compressing `"Lines 79-96"` into `"Lines 79-88"` ) to remove redundancy and improve clarity.

---

> ### Author Response · Authors · 2025-12-01
> **Author Responses to Reviewer cDMy- [Round2] (1/3)**
>
> We sincerely thank the reviewer for the positive feedback and the slight increase in the rating score. As suggested, we have  **added new `technical discussion` and `comparison experiments` with 3D editing**, and we would like to provide  further clarification to address your concerns:
>
>
>
>
> ## Point-to-Point Response
>
> > ### **Q1-1: As shown in Figure 1, the generation target is a combination of bread and candy.  However, the first stage generates 3D typography composed solely of bread, while the second stage introduces localized modifications to incorporate the candy style. Why was the candy style not integrated into the typography in the first stage, instead of being added as a localized control in the second stage?**
> >
> > **We apologize for the confusion caused by the original Figure 1, and we have largely revised Figure 1 to remove the ambiguity.** Our clarifications are as follows:
>
> **1. In the original Fig.1, the upper part "Global Generation" and the bottom part "Part-Level Control" are *two independent frameworks*, rather than two sequential stages of a single pipeline. In the revision, we have significantly revised Fig. 1 and removed `Global Generation` to eliminate this ambiguity.**
>
> In the original Fig.1, `"Part-Level Control"` essentially is another framework (**which is independent from `"Global Generation"`**). Specifically,  "Part-Level Control" first leverages `Glyph2Cloud` to generate component 3D point clouds for different parts, and then optimize 3D Gaussians via component-wise SDS, where DCA dynamically assigns component labels for 3D Gaussians. The pipeline is:
>
> - `[Inputs] Mask Components + Style Texts` -> `[Intermidate PC]Component 3D Point Clouds`-> `[Final Output]3DGS with Part-Level Styles`->`Done`
>
> *`"Part-Level Control"` does not receive any other inputs, such as the 3D representation of  `"Global Generation" `*.
>
> **2. `Localized modifications` and `3DGS optimization from point cloud` are performed *synchronously* (rather than sequentially); If sequentially, it would require a strategy to determine the editing orders ( especially for the beginning style),  which will largely impact the editing quality.**
>
> `"Part-Level Control"` **does not first generate a complete 3D representation and then apply local edits**. Instead, **`part-level style control` is integrated directly into the process of 3D optimization from scratch**. *If these steps were performed sequentially, one would need to devise a strategy for ordering the edits (especially for the beginning style), which is non-trivial and can lead to inconsistencies.*
>
> **3.  In the original Fig.1, ` 3D typography composed solely of bread` in the upper part has NO connection with the bottom part `a combination of bread and candy`. They have no connection.**
>
> In the original Fig. 1, it happens to generate the *same character* with *same bread style* purely by coincidence, **which unintentionally gave the impression that one is derived from the other.** In reality, **they are two completely independent generation procedures.**
>
> In the revised Fig. 1, to eliminate this ambiguity, we have changed a different character in `Glyph2Cloud` (from that of `Part-Level Control`).
>
> **4. `Global Generation` is a simplified version of `Part-Level Control`, i.e., One-Component Part-Level Control.**
>
>
> >
>
> > ### **Q1-2: The authors state that the first stage involves coarse-grained Gaussian points. However, the results presented in the first stage depict a refined 3D typography with a bread-style aesthetic, as shown in Figure 1. This level of detail is sufficient to incorporate other 3D editing techniques for localized control.**
> >
> > We apologize for the misunderstanding caused by the original Figure 1. Our clarifications are as follows:
>
> **1. In the original Fig.1, the upper part `"Global Generation" ` is NOT the first stage of our SplatFont3D, and its output 3D typography is NOT the input of the bottom part `"Part-Level Control"`.  They are *two independent frameworks, and both only receive `font masks` and `style texts` as inputs.***
>
> **2.  We perform `3DGS optimization from scratch` and `localized control` synchronously, rather than `first generating 3D typography` and `then performing local edits`. *Our ONE-pass 3D generation is easier, more efficient, and more effective than 3D Editing with TWO-pass 3D optimization***.
>
> We perform 3D Gaussian-splatting optimization from scratch with localized style controls integrated into the objective, rather than first producing a fine-grained 3D representation and then editing it (2nd-pass 3D optimization). *This synchronous formulation yields more consistent gradients, fewer artifacts, faster convergence, and a simpler, lower-overhead pipeline for exposing user-controlled local edits.* **In Appendix A.5, qualitative and quantitative analyses are given in Table 6 and Fig. 9.**
>
> **3. As shown in the *revised* Fig. 1, the initial 3D point cloud via `Glyph2Cloud` is very coarse and insufficient for 3D editing.**

---

> ### Author Response · Authors · 2025-12-01
> **Author Responses to Reviewer cDMy- [Round2] (2/3)**
>
> > ### **Q1: I don't doubt that this work is 3D generation. However, the second stage of this work is closely related to 3D editing techniques. Therefore, the lack of technical and experimental discussion on 3D editing diminishes the quality of the paper.**
> >
> > Thanks for your constructive suggestion, and as suggested, we have given an experimental discussion on `3D editing` in **Appendix A. 5** and a technical analysis in **Lines 161-167**. In short, our method belongs to `Part-Level Control from Scratch`, which is more effective and efficient than `3D Editing with local controls`. Our clarifications are as follows:
>
> **1. We perform `3DGS optimization from scratch` and `localized control` synchronously (in ONE-pass 3D optimization, i.e., Generation), rather than sequentially, as in the original Fig. 1; Instead, `3D editing` performs `3D generation` first and then `localized control` sequentially (in TWO-pass 3D optimization, i.e., Generation &  then Editing).**
>
> As our response to `Q1-1` & `Q1-2`, our SplatFont3D performs`3DGS optimization from scratch` and `localized control`  **synchronously**, where `localized control` can be regarded as **a kind of regularization during 3D generation**.  Instead, `3D editing` performs `3D generation` and `localized control` sequentially **with two-pass 3D optimization**, which first generates 3D assets and then performs local edits.
>
> **2. We agree that `3D editing with local controls` is a feasible approach, but our `part-level 3D generation from Scratch` is more effective and efficient than `3D editing` for 3D-AFG**. Technical advantages over 3D editing include:
>
> - **`"Direct 3D generation with single joint optimization"` is more efficient than `"3D editing with two-pass 3D optimization" (first generate and then edit)`, thus attaining faster  rendering speed and lower I/O overhead.**
>
> - **Most 3D editing approaches support only single-component or single-style edits per invocation. In contrast, our framework enables simultaneous control of multiple components with different styles in one pass.**
>
> - **3D editing requires precisely localizing part regions from 3D assets; however, this is difficult, especially for NeRF-based models and finer-grained editing on multiple components.**
>
> - **Generation from scratch is easier than editing existing 3D assets, attaining better visual quality with reduced  collapse and artifacts in edited regions, and better global consistency (than 3D editing)**
>
> Technical analysis is given in **Lines 161-167**.
>
> **3.  As suggested, we have added technical and experimental discussion on `3D editing` in Appendix A. 5.**
>
> **In Table 6,  we have provided a quantitative comparison with 3D editing models, where the initial 3D fonts for 3D editing are generated with our `SplatFont3D using global generation`.** In the table, `"SplatFont3D (Gen)"` refers to directly generating from scratch, and `"SplatFont3D (Edit)"` refers to firstly generating global 3D assets (with `Global Generation`) and then performing 3D editing with local edits.
>
> Results show that `3D editing` performs inferior to `3D generation from scratch`, including **lower visual quality & slower rendering speed**. **Qualitative comparison in Fig. 9 also indicates that `direct 3D generation` is more feasible than `3D editing` for 3D-AFG, especially for finer-grained controls on multiple parts.**
>
> | #  Strokes| Method |  CLIP$\uparrow$ | Alignment$\uparrow$ | Quality$\uparrow$ | V-LPIPS$\downarrow$ | V-CLIP$\uparrow$ |Speed (Min/R.)$\downarrow$|
> | :----------: |  :----------------: | -------: | :--------: | :--------: | :-------: | :-------: | :-------------: |
> |      3       |   GaussianEditor   |     0.71 |      2.87 |     12.89 |     0.29 |     0.87 |          29.16 |
> |              |     TIP-Editor     |     0.73 |      2.63 |     15.91 |     0.31 |     0.88 |          37.37 |
> |              | SplatFont3D (Edit) |     0.76 |      2.94 |     19.72 |     0.35 |     0.86 |          20.23 |
> |              | SplatFont3D (Gen)  | **0.84** |  **3.05** | **26.73** | **0.29** | **0.88** |      **13.99** |
> |||
> |      4       |   GaussianEditor   |     0.63 |      1.87 |     12.40 |     0.32 |     0.86 |          38.51 |
> |              |     TIP-Editor     |     0.69 |      1.65 |     19.09 |     0.33 |     0.86 |          60.38 |
> |              | SplatFont3D (Edit) |     0.70 |      1.89 |     23.14 |     0.37 |     0.85 |          24.09 |
> |              | SplatFont3D (Gen)  | **0.81** |  **2.50** | **31.89** | **0.31** | **0.87** |      **17.79** |
> |||
> |      5       |   GaussianEditor   |     0.63 |      3.04 |     19.74 |     0.34 |     0.85 |          45.99 |
> |              |     TIP-Editor     |     0.67 |      3.12 |     21.59 |     0.34 |     0.85 |          88.56 |
> |              | SplatFont3D (Edit) |     0.63 |      2.95 |     23.38 |     0.39 |     0.85 |          27.77 |
> |              | SplatFont3D (Gen)  | **0.77** |  **3.68** | **30.52** | **0.33** | **0.85** |      **21.42** |

---

> ### Author Response · Authors · 2025-12-01
> **Author Responses to Reviewer cDMy- [Round2] (3/3)**
>
> > ### **Q2: How is the process of "projecting these sampled 2D points into 3D space" implemented? Does it require depth information and camera parameters?**
> >
> > Thanks for your insightful comment, and we have added more details in **Fig.1** and **Lines 239-242**.  Specifically,
>
> Our projection process  **does NOT rely on depth maps or camera parameters**. Instead, we construct a coarse 3D point cloud directly from the 2D segmentation mask using a simple volumetric sampling strategy, i.e.,
>
> 1. **2D foreground sampling:**
>    We uniformly sample foreground pixels from the 2D segmentation map, forming a set of points lying on a single 2D plane.
> 2. **Depth-axis replication:**
>    We repeat this sampling operation along the depth axis (front-view) at fixed, uniformly spaced intervals. Each repetition produces a parallel 2D plane with the same foreground layout.
> 3. **Volumetric point cloud construction:**
>    Stacking these equally spaced 2D planes naturally forms a coarse 3D point cloud representing the volumetric shape of the glyph. This initialization is coarse and is only used as a starting geometry for the subsequent 3DGS optimization.
>
> Because **this procedure requires no camera calibration or depth estimation**, it does not rely on depth maps or camera parameters, and **the obtained initial 3D point cloud is very coarse (as shown in Fig. 1)**.
>
> >
>
> > ### **Q3：Has the motivation and principle behind "redesigning the sampling procedure through denoising intervention" been validated?**
> >
> > Such an idea has been thoroughly validated by many controlled image generation methods (based on diffusion models) [1-3]. Specifically,
>
> **1.  The part `"redesigning the sampling procedure through denoising intervention"` only influences the `2D Glyph Generation` and is not related to `"projecting these sampled 2D points into 3D space"` .**
>
> As we clarified in `Q-2`, ` "projecting these sampled 2D points into 3D space"` is achieved through `a simple volumetric sampling strategy`, and it is not related to `denoising intervention` for `2D Glyph Generation`.
>
> **2. Those methods for 2D image generation do not require re-training or fine-tuning the 2D prior diffusion models**, and **they can achieve controlled image generation simply through denoising intervention.**
>
>  For example,
>
> -  SEGA [1] can achieve `2D image generation with semantic control` by interacting with the diffusion process to flexibly steer it along semantic directions.  Such an approach requires no additional training or fine-tuning on the original 2D Stable-Diffusion.
>
> -  Literature [2] proposed `classifier-free diffusion guidance`, which guides diffusion models toward a conditioning signal through denoising intervention (i.e., mixing the conditional and unconditional score estimates during the diffusion generation process).
>
> -  A survey of `Image Editing with Diffusion Models` [3] also gives a detailed discussion about controlled image generation through denoising intervention (e.g., Fig. 6 in literature [3])
>
> [1] “SEGA: Instructing text-to-image models using semantic guidance". NeurIPS, 2023.
>
> [2] "Classifier-free diffusion guidance". *arXiv preprint arXiv:2207.12598*, 2022.
>
> [3] "Image Editing with Diffusion Models: A Survey." *arXiv preprint arXiv:2504.13226* , 2025.
>
>
>
> >
>
> ## Summary of our Revisions
>
> - We have **thoroughly revised Fig.1 and removed `Global Generation`** to eliminate the ambiguity. In addition,  `Global Generation` can be viewed as a special case of `Part-Level Control` , where only a single component is used.
> - As suggested, we have given **experimental discussion on `3D editing` in Appendix A. 5** (including quantitative and qualitative comparisons in **Table 6 and Fig. 9**) and **technical analysis in Related work Lines 160-166**.
> - As suggested, we have provided more details of `"projecting these sampled 2D points into 3D space"` in  **Fig. 1** and **Lines 239-242**.
>
> All modifications (in the 2nd round revision) are marked with the `red` color.

---

### Official Review · Reviewer_Wx5q · 2025-10-30

**Soundness:** 3
**Presentation:** 3
**Contribution:** 2
**Rating:** 6
**Confidence:** 4

**Summary:**

The paper addresses the unexplored direction of structured 3D artistic font generation (3D-AFG) by proposing SplatFont3D. Using 3D Gaussian Splatting (3DGS) as the representation and guided by Score Distillation Sampling (SDS) from 2D diffusion models, the method enables the generation of 3D artistic fonts from glyphs and textual style prompts, while also supporting part-level style control.

To tackle three core challenges—point cloud initialization, the lack of real 3D font data, and part drift and entanglement during training—the paper introduces three key designs:
	1.	Glyph2Cloud: Balances shape and style in the latent space of 2D diffusion to generate stylized glyphs. Foreground extraction is performed via text-guided segmentation, followed by point sampling to obtain a 3D point cloud for initializing 3DGS.
	2.	3DGS Optimization via SDS: Optimizes 3DGS by distilling supervision from a 2D diffusion model through the SDS paradigm, enabling the generation of high-quality 3D artistic fonts.
	3.	Dynamic Component Assignment (DCA): Dynamically updates the Gaussian points’ component labels during training based on component heatmaps from stylized 2D results, alleviating part entanglement caused by Gaussian drift and enabling explicit part-level style control.

Experiments and Results: The proposed method is compared with multiple baselines (DreamFusion, Latent-NeRF, MVDream, Wonder3D, Fantasia3D, GaussianDreamerPro, GsGen, DreamFont3D, etc.) under three settings—Global, Part-Level, and Global + Part-Level. Quantitative results demonstrate clear superiority in terms of quality and consistency metrics. Ablation studies further validate the effectiveness of the Glyph2Cloud and DCA designs.

**Strengths:**

By designing Glyph2Cloud and DCA specifically for font—a highly structured object—the paper effectively leverages the explicit representation advantages of 3D Gaussian Splatting (3DGS), such as high rendering efficiency and structural decomposability, making them well-suited for this task.
The work addresses a clear yet underexplored need in structured 3D artistic font generation, filling an important research gap in applying 3DGS to font modeling and generation.
The experiments are comprehensive, including multiple settings (Global, Part-Level, and Combined), extensive comparisons with baselines and ablations, and diverse qualitative and quantitative evaluations, demonstrating the robustness and effectiveness of the proposed approach.

**Weaknesses:**

Data and Generalization:
The evaluation dataset remains limited in scale and linguistic diversity. It is recommended to extend the experiments to fonts with higher stroke density, more complex structures, and additional languages. Moreover, it would be beneficial to report the control effectiveness and computational overhead under finer-grained component segmentation settings (e.g., more than three components).

Relation to Feed-Forward 3D Generation:
Recent feed-forward 3D generation methods (e.g., Trellis, UniLat3D) can efficiently produce 3DGS-based objects from 2D images. Although these approaches primarily target general object domains, it is suggested that the authors discuss their relevance—for instance, by comparing the efficiency and quality differences between generating 3D fonts from 2D artistic glyphs (x_g) using feed-forward methods and the proposed approach.

**Questions:**

1. Compared to directly generating 3D fonts from stylized 2D glyphs (x_g) using feed-forward methods such as Trellis, what specific advantages does the proposed approach offer?
2. Since the Dynamic Component Assignment (DCA) module relies on 2D label maps for guidance, how accurate are the reassigned component labels when severe occlusion or part overlap occurs from certain viewpoints?
3. What is the feasibility of extending the method to finer-grained (stroke-level) style control, and how would such extension impact optimization complexity and computational overhead?

---

> ### Author Response · Authors · 2025-11-20
> **Author Responses to Reviewer Wx5q (1/2)**
>
> We sincerely thank the reviewer for the positive evaluation of our work, and we have provided detailed point-by-point responses to address your concerns.
>
> ## Summary of our Revisions
>
> - As suggested, we have included the comparison with feed-forward methods Trellis, including the quantitative results in Table 1 and qualitative results in Fig.2, 10 & 11.
> - As suggested, in Section 4.3, we have extended SplatFont3D to finer-grained stroke-level style control, and have given the analysis of how such an extension would impact optimization complexity and computational overhead.
> - As suggested, in Appendix A.4,  we have extended SplatFont3D to  3D-AFG with more diverse styles and languages, as shown in Table 5 and Fig.8. It is worth noting that our evaluation of all methods on original data requires **~50 GPU-days**, and thus we can only compare with several SOTA methods due to the trade-off between cost and feasibility.
>
>
> ## Point-to-Point Response
>
> > ### **Q1: Compared to directly generating 3D fonts from stylized 2D glyphs (x_g) using feed-forward methods such as Trellis, what specific advantages does the proposed approach offer?**
> >
> > Thanks for your insightful question. We have conducted new experiments (including quantitative results in Table 1 and qualitative results in Fig.2, 10 & 11), and we found that **feed-forward methods such as Trellis struggle to generalize to rare and highly stylized domains like 3D-AFG, while our SplatFont3D attains better generalization to 3D-AFG**. Specifically,
>
> 1. **Limitations of feed-forward approaches: Struggle to generalize to rare domains like 3D-AFG.**
>
>     Trellis is trained directly on realistic 3D data and therefore preserves strong 3D consistency; however, 3D data is usually hard and expensive to obtain, and the scale of current 3D datasets is far smaller than 2D datasets (e.g., **Trellis with only 500K 3D samples  V.S. 2D Stable-Diffusion with billions of image-text pairs**). Thus, its performance is fundamentally constrained by the *limited scale and diversity of current 3D datasets*  when extending to rare generation domains (e.g., 3D-AFG, where 3D fonts are not common 3D objects). As shown in our qualitative comparison in Fig. 2, 10 & 11, Trellis often fails to synthesize high-quality 3D artistic glyphs due to this data scarcity.
>
> 1. **Advantages of our method:  Better generalization and Precise Part-level control.**
>
>    Our method leverages the rich visual priors of 2D Stable Diffusion (trained on orders of billions of image-text pairs) and optimizes a real-time 3D rendering via SDS. This combination enables our SplatFont3D to **generalize far better than Trellis to rarer 3D objects, especially for the rare 3D-AFG**. Moreover, our rendering pipeline is real-time and it supports **dynamic optimization**, allowing the geometry and appearance to adapt to the target style, whereas feed-forward models remain fixed after training.
>
> 1. **New quantitative and qualitative results: Our SplatFont3D achieves better performance than Trellis.**
>
>     We have conducted new quantitative experiments in Table 1 and qualitative experiments in Fig.2, 10 & 11), and the results show that our SplatFont3D outperforms Trellis.
>
> >
>
> > ### **Q2: How accurate are the reassigned component labels when severe occlusion or part overlap occurs from certain viewpoints?**
> >
> > Thanks for your good question. We demonstrate that component labels remain accurate, since the label reassignment solely depends on the front-view projection of the 3D representation. Specifically,
>
> - We always assign component labels of 3D-GS solely based on the **Front-View Projection**, i.e., *no matter what certain viewpoint is currently optimizing, DCA will reassign component labels of 3D representation based on its front-view projection*. This is because the front view provides the most semantically meaningful decomposition of components for 3D fonts. In contrast, component boundaries cannot be reliably identified from oblique or side-view projections, where strokes or structures may collapse into indistinguishable shapes.
>
> - Qualitative results in Fig. 2 show that: (1) our model is robust to *mild component-level overlap* (e.g., in the character *“四”*, where “八” overlaps with “口”); (2) It also handles *intra-component overlap* reasonably well (e.g., the internal structure “冬” in *“疼”*). These observations indicate that mild occlusion does not harm reassignment accuracy, though we avoid decomposing heavily occluded regions into separate components.
>
> - Moreover, in the front view, we prefer to choose component sets that are non-overlapping. If two components exhibit substantial overlap that makes separation ambiguous, we simply *merge them into a single component* to maintain labeling consistency and avoid introducing unreliable boundaries.

---

> ### Author Response · Authors · 2025-11-20
> **Author Responses to Reviewer Wx5q (2/2)**
>
> > ### **Q3: What is the feasibility of extending the method to finer-grained (stroke-level) style control, and how would such extension impact optimization complexity and computational overhead?**
> >
> > Thanks for your constructive suggestion. **As suggested, we have added the related experiments and analysis in Table 4  and Fig.6.** Specifically,
>
>
> **1. Our SplatFont3D can feasibly extend to stroke-level style control, since a stroke is also a component.**
>
> Since the stroke is a finer-grained version of the component,  our method is fully feasible to be extended to **finer-grained, stroke-level style control**. But it may increase the optimization complexity and computational costs.
>
> **2. New Results and Analysis: Impact on optimization complexity and computational overhead.**
>
> As suggested, we have conducted experiments to analyze how such stroke-level control (with higher component density) impacts optimization complexity and computational overhead. As shown in Table 4, we observe that higher component density results in higher computational costs and inferior visual quality due to increased optimization complexity.  We have also given qualitative results of stroke-level control in Fig. 6 in the revised manuscript.
>
> | # of Strokes  |  CLIP$\uparrow$ | Alignment$\uparrow$ | Quality$\uparrow$ | V-LPIPS$\downarrow$ | V-CLIP$\uparrow$ |GPU (GB) $\downarrow$| Speed (Min/R.)$\downarrow$|
> | :-----------: | ------------: | ----------: | ------------: | ----------: | ------------: | ---------: |---------: |
> |3|0.84|3.05|26.73|0.29|0.88|14.48|13.99|
> |4|0.81|2.50|31.89|0.31|0.87|16.14|17.79|
> |5|0.77|3.68|30.52|0.33|0.85|17.64|21.42|
> |6|0.80|2.69|34.85|0.34|0.84|19.24|25.06|
>
> >
>
> > ### **Q4: It is recommended to extend the experiments to fonts with higher stroke density, more complex structures, and additional languages.**
> >
> > Thanks for your constructive suggestion.  We have added more discussions and experiments to address your concerns.
>
> **1. New results on new styles and additional languages in Appendix A.4**
>
> We have extended our evaluation data with 2 additional languages `("Japanese" and "Korean")` with eight styles  `(“steel”, “gold”, “fire”, “white_cloud”, “rope”, “green_bamboo”, “tree_trunk”, “blood”)`. Therefore, our extended evaluation data covers **5 languages** `("numerals", "English", "Korean", "Japanese" and "Chinese")`, **16 different styles** for global, and $C_{16}^2$=**120** combinations for two-part.
>
>  **New experiments are conducted in Appendix A.4**. Considering that the original evaluation requires *1,760 renderings*  (e.g.,**~50 GPU-days for original evaluation**), we only compare SplatFont3D with several SoTA methods (covering Nerf, 3DGS, feed-forward based models)  for `"global style generation"` on the expanded data as shown in  **Table 5**, due to the trade-off between cost and feasibility. The **new qualitative results in  Fig. 8** also demonstrate that *our method can easily generalize to more diverse styles and languages for 3D-AFG, without requiring any real 3D data.*
>
> |     Methods     | CLIP$\uparrow$ | Alignment$\uparrow$ | Quality$\uparrow$ | V-LPIPS$\downarrow$ | V-CLIP$\uparrow$ |
> | :-------------: | ---: | --------: | ------: | ------: | -----: |
> |     Trellis     | 0.61 |      2.53 |   18.92 |    0.22 |   0.90 |
> | GaussianDreamer | 0.69 |      2.97 |   35.61 |    0.20 |   0.89 |
> |   DreamFont3D   | 0.77|      3.26 |   33.19 |    0.23 |   0.93 |
> |   SplatFont3D   | 0.79 |      3.70 | 46.20 | 0.21|  0.92|
>
>
> **2. More Experiments on Stroke-Level Control**
>
> Moreover, we also have extended experiments with higher component density in  **Table 4 and Fig. 6**, i.e., stroke-level control (which can refer to our response to your question **`Q3`**).
>
> **3. Our data is already the largest and most diverse in the domain of 3D-AFG**
>
> Importantly, in the domain of 3D font generation, our data is **substantially larger** than prior work. For example, **DreamFont3D (SIGGRAPH ‘24)** crafts only **50 font mask images** across English, Chinese, numerals, and symbols. In contrast, our evaluation data attains much larger scales, making our evaluation the largest and most diverse one in this area.
>
> We hope that our experiments can provide considerable evidence of the robustness and cross-language generalization of our SplatFont3D for 3D-AFG.

---

### Official Review · Reviewer_rrgN · 2025-11-01

**Soundness:** 2
**Presentation:** 3
**Contribution:** 2
**Rating:** 4
**Confidence:** 3

**Summary:**

This paper introduces SplatFont3D, a novel framework for generating 3D artistic fonts from text prompts and 2D font image, emphasizing structure-aware synthesis and precise part-level style control. Key challenges include maintaining semantic and structural constraints of fonts, achieving fine-grained part-level stylization, and overcoming the scarcity of 3D font data. They also propose a dynamic component assignment strategy to handle Gaussian drift and enable part-level control.

**Strengths:**

1. **Reasonable Solutions towards the 3D fonts challenge**: They use 2D diffusion priors to initialize 3D point clouds, balancing shape preservation and stylistic fidelity through denoising interventions and segmentation. The dynamic component assignment is a smart solution to Gaussian drift, enabling explicit part decomposition superior to implicit representations like NeRF.
2. **Extensive evaluation results**: The evaluation is comprehensive, covering global and part-level scenarios across diverse characters and styles. Diverse metrics like CLIP score, Alignment, Quality, V-LPIPS, and V-CLIP. Comparisons with state-of-the-art baselines demonstrate empirical superiority.

**Weaknesses:**

1. **Limited Scope of Evaluation Data**: The dataset comprises only 44 characters with 2 styles and modes each (1760 pairs total), which may not fully represent the diversity of fonts or languages. While including Chinese characters adds some breadth, the focus on limited categories (e.g., fruits, foods) could bias results toward simpler styles, potentially limiting generalizability to more complex or abstract prompts.
2. **Lack of ablation studies about each component**. The paper lacks several important ablation studies to prove the effectiveness and necessity of the proposed components. For example how to prove the dynamic component assignment is necessary. Also the reviewer thinks the most critical part is the initialization module. Can the authors show how is the comparison if the initializations of all the methods are the same?
3. While the challenges remain, the qualitative results could still be improved.
4. The method avoids collecting real 3D data, which is a strength, but how does it compare to supervised fine-tuning on synthetic 3D fonts? Would incorporating even a small 3D dataset further improve performance, and if so, under what conditions?

**Questions:**

Please refer to Weaknesses part.

**Details Of Ethics Concerns:**

None.

---

> ### Author Response · Authors · 2025-11-20
> **Author Responses to Reviewer rrgN (1/2)**
>
> We appreciate the reviewer's recognition of the significance of our work, and we have provided detailed point-by-point responses to address your concerns.
>
> ## Summary of our Revisions
> - As suggested, in section 4.3, we have shown the comparison if the initializations of all the methods are the same.
> - As suggested, in Appendix A.4,  we have extended our SplatFont3D to 3D-AFG with more diverse styles and languages, as shown in Table 5 and Fig. 8. It is worth noting that:
>   - Our evaluation of all methods on original data requires **~50 GPU-days**, and thus we can only compare with several SOTA methods due to the trade-off between cost and feasibility.
>   - Our original data is already the largest and most diverse in the domain of 3D-AFG.
>
>
> ## Point-to-Point Response
>
> >
> > ### **Q1: Limited Scope of Evaluation Data.**
> >
> >We agree with the reviewer, and we have given new results and discussions to address your concerns:
>
> 1. **It is the trade-off between Cost and Feasibility, and Larger data will not affect the conclusion**
>
>    Most of the current 3D-aware methods rely on real-time rendering, making generation time scale linearly with dataset size.
>
>    - In our benchmark, we already evaluate *10 state-of-the-art models*, whose rendering speeds range from *102 minutes/sample*  (GaussianDreamPro) to *14.3 minutes/sample* (GaussianDreamer) as listed in Fig. 3. Given that our evaluation requires 1,760 renderings  (e.g., **~50 GPU-days for current evaluation**), further enlarging the dataset would incur an enormous computational cost.
>
>    - *We believe that expanding the dataset would therefore be prohibitively expensive, while not affecting the conclusions*: performance gaps between methods are **large and stable**, and remain consistent across additional internal subsets we tested. Hence, the current dataset size represents a practical and feasible trade-off that still supports the validity of our comparisons.
>
> 2. **Our original data is already the largest and most diverse in the domain of 3D-AFG**
>
>    In the domain of 3D font generation, our data collection is *substantially larger* than prior work. For example, **DreamFont3D (SIGGRAPH ‘24)** crafts only **50 font mask images** across English, Chinese, numerals, and symbols. In contrast, our evaluation data attains much larger scales, making our evaluation be the largest and most diverse one in this area. So far, there were very few other works for 3D-AFG.
>
> 3. **New Results with More Styles and Languages**
>
>     To further address the reviewer’s concerns about diversity, we additionally evaluate our method on **eight new visual styles** and **two more languages** (**Korean**  and  **Japanese**). Our extended evaluation data covers **5 languages** `("numerals", "English", "Korean", "Japanese" and "Chinese")`, **16 different styles** for global, and $C_{16}^2$=**120** combinations for two-part.
>     **New results on extended evaluation data are given in Appendix A4.** Due to the trade-off between cost and feasibility, we only compare SplatFont3D with several SOTA methods for `"global style generation"` as shown in Table 5. The results show that our method generalizes well across these new settings and outperforms previous SOTA.  The **new qualitative results in Fig. 8** also demonstrate that *our method can easily generalize to more diverse styles and languages for 3D-AFG, without requiring any real 3D data.*
>
>     |     Methods     | CLIP$\uparrow$ | Alignment$\uparrow$ | Quality$\uparrow$ | V-LPIPS$\downarrow$ | V-CLIP$\uparrow$ |
>     | :-------------: | ---: | --------: | ------: | ------: | -----: |
>     |     Trellis     | 0.61 |      2.53 |   18.92 |    0.22 |   0.90 |
>     | GaussianDreamer | 0.69 |      2.97 |   35.61 |    0.20 |   0.89 |
>     |   DreamFont3D   | 0.77 |      3.26 |   33.19 |    0.23 |   0.93 |
>     |   SplatFont3D   | 0.79 |     3.70 |   46.20 |   0.21|  0.92 |
>
> >
>
> > ### **Q2-1:** **Lack of ablation studies about each component**.
> >
> > Thanks for your constructive comments, and we have conducted the important ablation studies (in **Section 4.3**, **Table 2**, **Figure 4**, and **Figure 5**) to demonstrate the effectiveness and necessity of the proposed components (including Dynamic Component Assignment **DCA**, and Initialization Module Glyph2Cloud **G2C**). Specifically,
>
> 1. **Quantitative Ablation Results on Each Component in Table 2.**
>
>    We presented quantitative ablation results in Table 2, where ablation results demonstrate the effectiveness of G2C and DCA of SplatFont3D.
>
> 2. **Qualitative Ablation Results in Fig. 4**
>
>    Fig. 4 shows the qualitative ablation results of DCA and G2C, and both module contributes significantly to the final rendering performance.
>
>  3. **G2C for Shape-Style Tradeoffs in Fig. 5**
>
>     Fig. 5 shows that G2C can help achieve customized shape-style tradeoffs for 3D-AFG by dynamically adjusting the hyperparameters *K* and *α* in Eq. (4).

---

> ### Author Response · Authors · 2025-11-20
> **Author Responses to Reviewer rrgN (2/2)**
>
> > ### **Q2-2:** **Can the authors show how is the comparison if the initializations of all the methods are the same?**
> >
> > As suggested, we have given new experiments in Table 3 to address your concerns, specifically,
>
> **1. Most 3D baselines are text-to-3D models that receive only texts as inputs, thus making our G2C inapplicable.**
>
> - A unified initialization is **not feasible** for most existing 3D generative models, because the majority of text-to-3D approaches (especially NeRF-based ones) take only texts as inputs and **do not support point-cloud or image initialization**.
> - Only 3DGS-based approaches require an initial point cloud during intermediate stages, and the core contributions of different 3DGS methods are how to initialize the 3D point cloud  (e.g., GaussianDreamer vs. GaussianDreamerPro vs. Our SplatFont3D). In addition, feedforward models may receive images as inputs, and thus our G2C can be applied.
>
> **2. Comparison with Compatible Methods using Same Initializations**
>
> Therefore, in Table 3, we evaluate only the subset of models for global generation that **do** rely on initial point clouds or images by conducting the same initialization (e.g., either using G2C or not). We can observe that our G2C module improves the performance of most 3D models for 3D-AFG, and our SplatFont3D achieves the best results.
>
> |     Methods   |G2C |  CLIP$\uparrow$ | Alignment$\uparrow$ | Quality$\uparrow$ | V-LPIPS$\downarrow$ | V-CLIP$\uparrow$|
> | :-------------: |  :-------------: |---: | --------: | ------: | ------: | -----: |
> |     Wonder3D     | X  | 0.64 | 3.09 | 25.28 | 0.51 | 0.74 |
> ||O|0.78|3.94|32.48|0.38|0.80|
> |     Trellis     | X  | 0.61 | 2.85 | 20.64 | 0.20 | 0.91 |
> ||O|0.72|2.39|29.01|0.21|0.91|
> | GaussianDreamer |X | 0.71 | 3.62 | 40.36 | 0.19 | 0.92 |
> ||O|0.70|3.17|35.15|**0.17**|0.90|
> |   SplatFont3D (Ours)   | X  | 0.72 | 3.68 | 32.79 | 0.24 |0.87|
> ||O|**0.80**|**4.02**|**53.11**|$\underline{0.18}$|**0.93**|
>
>
> >
>
> > ### **Q3:** **Qualitative results could still be improved**.
>
> > We thank the reviewer for the thoughtful comment.
>
> - We agree that this line of research 3D-AFG is still at an early stage and that further improvements (especially for qualitative results) would be valuable.
> - Even so, our current approach already achieves state-of-the-art results, outperforming existing baselines for 3D-AFG (according to our quantitative and qualitative results in Table 1 and Fig.2). The improvements brought by C2C and DCA have been carefully evaluated, and *we have also achieved structure-ware 3D-AFG with part-level control, which is rarely explored for 3D-AFG before.*
>
>
> >
>
> > ### **Q4:** **How does it compare to supervised fine-tuning on synthetic 3D fonts. Would incorporating even a small 3D dataset further improve performance, and if so, under what conditions?**
> >
> > Thanks for your insightful and constructive comment. We agree that, in principle,  training a model directly on a **large-scale 3D dataset** preserves the strongest 3D consistency; In contrast, *a small* 3D dataset would introduce bias and hurt generalization, for the following reasons:
>
> 1. **Our method attains better generalization by leveraging large-scale pretrained 2D priors.**
>
>     Our method benefits from the strong visual prior of large 2D diffusion models (e.g., Stable Diffusion pre-trained with *billions of image-text pairs*), enabling 3D artistic glyph generation **without any 3D supervision**. This 2D prior brings rich knowledge of shapes, materials, and artistic styles, which in turn provides considerable generalization across unseen glyphs and styles.
>
> 2. **Small 3D datasets would introduce bias and hurt generalization.**
>
>     Training or fine-tuning on a *small* set of synthetic or real 3D fonts typically leads to overfitting toward the limited observed shapes and stylistic patterns. This has been consistently observed in Machine Learning that the model becomes biased toward seen training data, and thus it loses the ability to generalize to new styles or unseen characters, which is the core goal and strength of our setting.
>
> 3. **Large-scale 3D data would help—but is currently unavailable.**
>
>     We agree that a sufficiently large and diverse 3D artistic glyph dataset would preserve strong 3D consistency. However, to the best of our knowledge, no such 3D artistic font dataset is publicly available, and manually producing one would incur prohibitive human cost. Moreover, constructing such a dataset would need to contain **tens of thousands of professionally designed 3D glyph instances with expert designers**, and also, the simple 3D font generation (via synthetic algorithms based on simple rules) would not offer the required artistic quality, textural richness, or style diversity.

---

### Author Response · Authors · 2025-12-02
**Summary Clarification for Area Chair**

Dear Area Chair,

We fully understand the significant workload you are facing under such unexpected and challenging circumstances, and we sincerely appreciate your valuable time and insightful efforts in thoroughly evaluating our submission and rebuttal.  Our summary is as follows:

---

> ### **0. Core Values and Significance of our work**

- **Accessible part-level 3D artistic font design**:

  Creating 3D artistic fonts manually is highly challenging, as it requires specialized expertise in both artistic design and 3D modeling. However, our work allows **any user to create customized 3D artistic fonts with part-level control**, ***with no need for professional skills***.

-  **Filling an important unexplored research gap**:

    Previous research primarily focused on 2D font generation, while **3D artistic font generation (3D-AFG) remains largely unexplored**. Our work fills this important gap.

- **Attaining core technical advantages**:

  - ***Data-free 3D creation***:  No real 3D data required, and we achieve part-level 3D font generation **without ground-truth supervision**.
  - ***Strong generalization***: By leveraging 2D priors of  StableDiffusion, our approach generalizes to **more universal settings of unseen characters and artistic styles**.
  - ***Faster rendering speed***: Inherits high efficiency of 3DGS and surpasses most existing 3D baselines.
  - ***Explicit part-level control***: Explcit component-wise control via 3DGS, unlike NeRF’s implicit representations.
  - ***SOTA performance***: Outperforms existing 3D baselines.

- **All reviewers consistently affirmed the value, novelty, and contributions of our work**, including:

  - Reviewer **`Wx5q`**：`"The work addresses a clear yet underexplored need in structured 3D artistic font generation, filling an important research gap in applying 3DGS to font modeling and generation." `

  - Reviewer **`rrgN`**： `"Reasonable Solutions towards the 3D fonts challenge"`, `"Comparisons with state-of-the-art baselines demonstrate empirical superiority"`

  - Reviewer **`cDMy`**：`"The paper successfully achieves a novel typography generation task"`,  `"SplatFont3D has significant advantages in generation quality, controllability, and training efficiency."`, `"Although I maintain my opinion on technical pipeline, this does not undermine the main contributions of the paper."`

---

> ### **1. Ratings:**

- **Initial Ratings (& Confidence):**  6 (4), 4 (3), **2** (4)
- **Ratings Before Privacy-Leak:** 6 (4), 4 (3), **4** (4)

  (P.S. Only **ONE**  reviewer responded to us during the rebuttal discussion.)

> ### **2. Brief of Rebuttal Responses from Reviewers:**

   - **Reviewer** **`cDMy`**:   **`Reject 2 (Conf. 4)` $\rightarrow$ `Borderline-Reject 4 (Conf. 4)`  showed a clear positive shift in attitude.**
     - **Recognition after initial clarification**: `"Although I maintain my opinion on technical pipline, this does not undermine the main contributions of the paper."`
     - **Why slightly increase rating (2$\rightarrow$4):** `"Given the authors' improvements to the existing issues and their partial clarification of my questions, I am willing to slightly increase my score."`
     - **Look forward to further clarification**:  `"I still have some differing opinions and look forward to further clarification from the author."`

- **Reviewer `Wx5q`**  & **`rrgN` have not yet responded before privacy-leak accident**  ：

   We regret that Reviewers  **`Wx5q`** and **`rrgN`** have not yet responded before privacy-leak, but we have provided **point-to-point responses with new analysis and experiments** to address their concerns.

> ### **3. Major Concerns & Responses**

| **Major Concerns**      |**Response & Solution** |
| :- | :-|
|**Comparison with 3D Editing**| **(1)** We provided **experimental comparison with `3D editing` in Appendix A. 5, Table 6, and Fig. 9** and **technical analysis in Related Work Lines 160-166**. |
||**(2)** Comparison show that **our generation from scratch is more effective and efficient than 3D editing.**|
|||
| **Evaluation Data Scale; More Languages and Styles** | **(1)** Extended results on **New Styles and More Languages in Appendix A.4**, *spanning 5 languages, 16+ styles, their combinatorial variants*. |
||**(2)** Notably, **our evaluation data is already the largest and most diverse in the domain of 3D-AFG**, substantially larger than prior attempt  (i.e., **DreamFont3D with only 50 samples**). ***Moreover, evaluations of all methods on our original data require ~50 GPU-days***|
|||
|**Extension to Stroke-Level Style Control and Its Impacts**| We provided **new experiments and analysis for  `Stroke-Level Style Control`  in Table 4 and Fig.6.** |
|||
|**Ablation Studies about Each Component.**|We proveided **more detailed ablation study  in Section 4.3**.|

> ---

Finally, we respectfully defer to the Area Chair’s expert judgment, based on **our revisions and detailed point-by-point responses to reviewers**.

---

### Meta-Review · Area_Chair_MA7D · 2026-01-07

**Summary:**

The paper proposes SplatFont3D, a system for Text-to-3D Artistic Font Generation. An image-based model first converts 2D glyph parts to point clouds, and then an SDS-based loss is used to optimize a 3D Gaussian Splatting.

Major concerns are (1) limited evaluation data, (2) the complexity of the results is relatively easy, and (3) the pipeline is a bit of engineering and lacks novelty. Based on these concerns, the AC decided to reject the paper. The authors are encouraged to incorporate feedback from the reviewers and resubmit the paper to a future venue.

**Reviewer Concerns:**

Major concerns are (1) limited evaluation data, (2) the complexity of the results is relatively easy, and (3) the pipeline is a bit of engineering and lacks novelty.

**Reviewer Scores:**

4, 6, 2 -> 4, 6, 2

---

### Decision · Program_Chairs · 2026-01-26

Reject